

# A microfluidic approach for sequential assembly of siRNA polyplexes with a defined structure—activity relationship

Dominik M. Loy[1], Philipp M. Klein[1], Rafał Krzysztoń[2,3],
Ulrich Lächelt[1], Joachim O. Rädler[2,3] and Ernst Wagner[1]

[1] Department of Pharmacy, Ludwig-Maximilians-Universität München, Munich, Bavaria, Germany
[2] Faculty of Physics, Ludwig-Maximilians-Universität München, Munich, Bavaria, Germany
[3] Graduate School of Quantitative Biosciences (QBM), Ludwig-Maximilians-Universität München, Munich, Bavaria, Germany

Corresponding authors
Dominik M. Loy,
dominik.loy@cup.uni-muenchen.de
Ernst Wagner,
ernst.wagner@cup.uni-muenchen.de

## ABSTRACT

Therapeutic nucleic acids provide versatile treatment options for hereditary or acquired diseases. Ionic complexes with basic polymers are frequently used to facilitate nucleic acid's transport to intracellular target sites. Usually, these polyplexes are prepared manually by mixing two components: polyanionic nucleic acids and polycations. However, parameters such as internal structure, size, polydispersity and surface charge of the complexes sensitively affect pharmaceutical efficiency. Hence a controlled assembly is of paramount importance in order to ensure high product quality. In the current study, we present a microfluidic platform for controlled, sequential formulation of polyplexes. We use oligo-amidoamines (termed "oligomers") with precise molecular weight and defined structure due to their solid phase supported synthesis. The assembly of the polyplexes was performed in a microfluidic chip in two steps employing a design of two successive Y junctions: first, siRNA and core oligomers were assembled into core polyplexes. These core oligomers possess compacting, stabilizing, and endosomal escape mediating motifs. Second, new functional motifs were mixed to the core particles and integrated into the core polyplex. The iterative assembly formed multi-component polyplexes in a highly controlled manner and enabled us to investigate structure-function relationships. We chose nanoparticle shielding polyethylene glycol (PEG) and cell targeting folic acid (termed "PEG-ligands") as functional components. The PEG-ligands were coupled to lipid anchor oligomers via strain promoted azide-alkyne click chemistry. The lipid anchors feature four cholanic acids for inserting various PEG-ligands into the core polyplex by non-covalent hydrophobic interactions. These core-lipid anchor—PEG-ligand polyplexes containing folate as cell binding ligand were used to determine the optimal PEG-ligand length for transfecting folate receptor-expressing KB cells in vitro. We found that polyplexes with 20 mol % PEG-ligands (relative to $n_{core\ oligomer}$) showed optimal siRNA mediated gene knock-down when containing defined PEG domains of in sum 24 and 36 ethylene oxide repetitions, 12 EOs each from the lipid anchor and 12 or 24 EOs from the PEG-ligand, respectively. These results confirm that transfection efficiency depends on the linker length and stoichiometry and are consistent with previous findings using core—PEG-ligand polyplexes formed by click modification of

azide-containing core polyplexes with aforementioned PEG-ligands. Hence, successive microfluidic assembly might be a potentially powerful route to create defined multi-component polyplexes with reduced batch-to-batch variability.

## INTRODUCTION

Together with deepened understanding of molecular pathways in hereditary and acquired human diseases comes a growing field of possible applications for nucleic acid-based drugs (*Shi et al., 2016*). Recent clinical trials have hinted at the therapeutic potential of non-viral gene carriers (*Sardh et al., 2018*; *Titze-de-Almeida, David & Titze-de-Almeida, 2017*), culminating in the approval of patisiran (*Haussecker, 2018*; *Adams et al., 2018*). Payloads of these synthetic carriers are usually various nucleic acids that are condensed into particles by cationic lipids (*Kulkarni, Cullis & Van der Meel, 2018*) or polycations (*Lächelt & Wagner, 2015*). These polycations have been the focus of extensive research in the past and have been continuously improved to find the optimal balance between various properties, for example compaction, intracellular release (*Leong & Grigsby, 2010*), cell uptake, serum stability, and toxicity (*Hall et al., 2017*). Formulation, however, is only gradually seen to be of importance as well (*Valencia et al., 2012*). Analogous to approval procedures of protein-based drugs, manufacturing processes are an integral part of the product and therefore control over them is critical (*Wilhelm et al., 2016*). Polyplexes (*Felgner et al., 1997*) are often prepared by batch wise mixing polycations with nucleic acids either by vigorous pipetting or shaking. Although this method is convenient, particle formation is kinetically controlled and charge neutralization in polyplexes occurs in around 50 ms (*Braun et al., 2005*). Consequently, limited batch-to-batch reproducibility inevitably leads to variable particle properties, which in turn complicate the establishment of precise structure—function relationships. Size and shape, for example, play a major role in deciding the uptake route into cells (*Rejman et al., 2004*; *Sykes et al., 2014*). They are, however, heavily influenced by assembly conditions. Moreover, each additional component included in the formulation complicates the preparation of defined nanoparticles. Generally, there are two distinct approaches to standardized nanoparticle production. The top-down process, on the one side, produces particles from larger materials, for example with the PRINT method developed by *Rolland et al. (2005)*. The advantage of this approach is a high control over size and shape of printed particles, although particle purification can be difficult. The bottom-up process, on the other side, produces particles from smaller building units or starting materials which assemble into bigger objects. Here, particle size and shape are controlled during the assembly process and additionally influenced by the design of the educts. In case of ionic polyplexes, the self-assembly process is based on electrostatic interaction between oppositely charged materials. However, control over size and shape is challenging. A widely used macrofluidic

approach to standardize nanoparticle production is the application of a T-junction. It enables the continuous production of large quantities of, for example, lipoplexes (*Clement et al., 2005*) or polyplexes (*Kasper et al., 2011*) with a turbulent mixing regime. Microfluidic approaches (*Liu et al., 2017*) to the bottom-up production of polyplexes can be broadly divided in droplet- (*Seemann et al., 2012*) and hydrodynamic focussing- (*Lee et al., 2016*) based systems. Both methods are suitable, since polyplex production is performed in aqueous systems and requires fast reaction times. Emulsion based systems have the advantage of discrete reaction chambers with picolitre volumes, but they are usually unstable and need additional surfactants and oily phases to stabilize droplets (*Ho et al., 2011*). Laminar flow-based systems have the advantage of producing carriers continuously while mixing of reactants is diffusion controlled only. Mixing speeds can be manipulated by employing baffle structures, organic solvents or external energy sources to influence the time scales reactants need to reach their counterparts allowing for a greater control over particle properties. It has been shown in previous studies that microfluidic-based assembly improves physicochemical properties of produced particles (*Belliveau et al., 2012*; *Grigsby et al., 2013*; *Koh et al., 2009*). Besides control over the assembly process, control over the precise structure of nanoparticle's components is essential as well. Solid phase supported synthesis (SPSS) of sequence defined oligo(ethanamino)amides (*Schaffert et al., 2011*) in our lab has the potential to integrate any functional element at any place in the oligomer's structure. The crucial parameter is the biological performance of polyplexes assembled from these oligomers. We have identified key units in the oligomer's structure: polycationic succinoyl tetra-ethylene pentamine (Stp) units for complexing nucleic acids and tyrosines and fatty acids for stabilizing (*Fröhlich et al., 2012*; *Troiber et al., 2013a*) the resulting nanoparticle. Usually, additional chemical moieties, for example, for shielding the nanoparticle and targeting (*Klein et al., 2018*) certain receptors, are integrated into the oligomer's structure to increase biological performance. These additional units, albeit required for efficient nucleic acid delivery, can alter polyplex formation processes (*Freund et al., 2018*). We set out to combine the advantages of sequence defined oligomers with enhanced control over the formulation process in order to generate precise multi-component polyplexes. To this end, we have used a laminar flow-based micromixer to generate polyplexes continuously. We show that the production of three and four component polyplexes is feasible. These polyplexes are assembled on-chip from siRNA, cationic core oligomers (*CO*), and polyethylene glycol (PEG)-ligands with zero to 48 ethylene oxide (EO) repetitions, which are integrated non-covalently into core polyplexes by lipid anchors containing an additional 12 EO repetitions. We use this system to identify the best PEG-ligand length for transfecting KB cells. Additionally, we compared and tested the same set of PEG-ligands on previously published two component polyplexes.

## MATERIALS AND METHODS

### Materials

Main suppliers: Biochrom (Biochrom GmbH, Berlin, Germany), Iris (Iris Biotech GmbH, Marktredwitz, Germany), Promega (Promega GmbH, Mannheim, Germany), Roth (Carl Roth GmbH + Co. KG, Karlsruhe, Germany), Sigma (Sigma-Aldrich Chemie GmbH,

Munich, Germany, now part of Merck KGaA, Darmstadt, Germany), Thermo (Thermo Fisher Scientific GmbH, Schwerte, Germany), VWR (VWR International GmbH, Darmstadt, Germany).

Solvents: Purified water (produced with Ultra Clear® GP UV UF, Evoqua Water Technologies GmbH, Günzburg, Germany), acetone *HPLC grade* (VWR), dichloromethane *ACS reagent* (DCM, Bernd Kraft GmbH, Duisburg), dimethylformamide *peptide grade* (DMF, Iris), dimethyl sulfoxide *for synthesis* (dimethylsulfoxide (DMSO), Acros Organics, Geel, Belgium), N-methyl pyrrolidon *peptide grade* (NMP, Iris), acetonitrile *for HPLC* (VWR), methanol *for HPLC* (VWR), ethanol *Ph.Eur.* (VWR), piperidine *peptide grade* (Iris), pyridine *puriss. p.a.* (Acros Organics, Geel, Belgium), n-hexane *puriss. p.a.* (VWR), tert-butylmethylether *for synthesis* (tBME; VWR).

Chemicals: 4-(2-hydroxyethyl)-1-piperazineethanesulfonic acid *ultra-pure* (HEPES, Biomol GmbH, Hamburg, Germany), D(+)glucose monohydrate *DAB* (Loewe Biochemica GmbH, Sauerlach, Germany), NaOH pellets *puriss.* (VWR), NaOH 1M standard solution (Thermo), HCl 1M standard solution (VWR), glycylglycine, ≥ 99% (Sigma), $MgCl_2 \times 6\,H_2O$, *p.a.* (Merck), ethylenediaminetetraacetic acid-$Na_2 \times 2H_2O$ ≥ 99% (EDTA, Sigma), DL-dithiothreitol ≥ 98% (DTT, Sigma), 5′-ATP-$K_2$ ≥ 92% (Sigma), coenzyme A $Li_3$ ≥ 93% (Sigma), glycerol ≥ 86% *Ph.Eur.* (Roth), bromophenol blue *ACS reagent* (Sigma), tris base ≥ 96% (Sigma), boric acid ≥ 99.5% (Sigma), KCl *p.a.* (Sigma), $Na_2HPO_4$ *p.a.* (Merck), $KH_2PO_4$ *p.a.* (Merck), protected Fmoc-α-amino acids (Iris), Fmoc-N-amido-dPEG12-acid (discrete PEG), Fmoc-N-amido-dPEG24-acid (Quanta Biodesign, Powell, OH, USA), N10-(tri- fluoroacetyl)pteroic acid (Clauson-Kass A/S, Farum, Denmark), Fmoc-Glu-O-2-PhiPr (VWR), 2-chlorotrityl chloride resin (200–400 mesh, 1% DVB crosslinking, Iris), ammonia solution 25% *Ph.Eur.* (Roth), N,N-diisopropylethylamine (DIPEA, Iris), trifluoroacetic acid *peptide grade* (TFA, Iris), triisopropylsilane ≥ 98% (TIS, Sigma), phenol *p.a.* (AppliChem GmbH, Darmstadt, Germany), KCN *ACS reagent* (Sigma), acetic anhydride ≥ 99% *puriss. p.a.* (Sigma), benzotriazol-1-yl-oxytripyrrolidinophosphonium hexafluorophosphate (PyBOP, MultiSynthech, Witten, Germany), 1-hydroxybenzotriazole hydrate ≥ 97% (HOBt, Sigma), 5β cholanic acid (CholA) ≥ 99% (Sigma), sephadex G10 (VWR), agarose *BioReagent* (Sigma), Sylgard® 184 polydimethylsiloxane silicon elastomer base (PDMS, Dow Corning GmbH, Wiesbaden), PDMS curing agent (Dow Corning GmbH, Wiesbaden), uranyl formate, Dibenzocyclooctyne (DBCO)-PEG4-N-hydroxysuccinimidyl ester ≥ 95% (DBCO, Sigma), tri-chloro(1H,1H,2H,2H-perfluorooctyl)silane ≥ 97% (Sigma).

Dyes: Ethidium bromide (EtBr) 10 mg/ml in $H_2O$ (Sigma), 3-(4,5-dimethylthiazol-2-yl)-2,5-diphenyltetrazolium bromide (MTT, Carl Roth), DBCO-PEG4-Atto488 (Jena Bioscience GmbH, Jena, Germany), GelRed™ 10000× in $H_2O$ (VWR), ninhydrin ≥ 95% *purum* (Sigma).

Nucleic acids: siGFP sense: 5′-AuAucAuGGccGAcAAGcAdTsdT-3′, antisense: 5′-UGCUUGUCGGCcAUGAuAUdTsdT-3′ (Roche Molecular Systems, Inc, Pleasanton, CA, USA), siAha1-Cyanine 5 (Cy5) sense: 5′-(Cy5-NHC6)-GGAuGAAGuGGAGAuuAGudTsdT-3′, antisense: 5′-ACuAAUCUCcACUUcAUCCdTsdT-3′ (Roche Molecular Systems, Inc, Pleasanton, CA, USA), siCtrl sense: 5′-AuGuAuuGGccuGuAuuAGdTsdT-3′, antisense:

5′-CuAAuAcAGGCcAAuAcAUdTsdT-3′ (Roche Molecular Systems, Inc, Pleasanton, CA, USA). Small letters: 2′ methoxy; s: phosphorothioate.

Cell culture: 100× benzylpenicillin sodium (10000 E) + streptomycinsulfate (10 mg/ml) solution (Biochrom), Gibco™ fetal bovine serum, (FBS, Thermo), RPMI 1640 (R2405-500ML, Sigma), RPMI 1640, folate free (27016021, Thermo), 10× trypsin/EDTA in PBS (Biochrom), collagen A, 0.1% in HCl, one mg/ml (Biochrom), heparin sodium 25k (Ratiopharm, Ulm, Germany), cell culture 5× lysis buffer (Promega), VivoGlo™ D-luciferin potassium (Promega).

Further materials: Microreactors (MultiSynTech, Witten, Germany), carbon coated copper grids (Ted Pella, Inc., Redding, CA, USA, 300 mesh, 3.0 mm O. D.), 96 well plates (TPP 92096; Faust Lab Science GmbH, Klettgau, Germany), cell culture flasks (TPP90075; Faust Lab Science GmbH, Klettgau, Germany), Versilon™-Inert-Schlauch SE-200, 1.6 × 3.2 mm, Wd 0.8 mm, PP-T-Tüllenverbinder 1.6 mm, PP-Luer connector, female, PP-Luer connector, male (ProLiquid GmbH, Überlingen, Germany), Hamilton syringes: syr one ml 1001 TLL, $d_{inner}$ = 4.61 mm, syr 500 μl 1750 TLL-XL, $d_{inner}$ = 3.26 mm, syr 100 μl 1710 TLL-XL, $d_{inner}$ = 1.46 mm, needles: NDL ga27, 90 mm, pst4 (Hamilton Bonaduz AG, Bonaduz, Switzerland), syringe pumps: LA-122, LA-120, LA-160 (Landgraf Laborsysteme HLL GmbH, Langenhagen, Germany), LabView 2017 (National Instruments, Austin, TX, USA), biopsy puncher (World precision instruments; ID = 0.96 mm; OD = 1.26 mm).

## Oligomer synthesis

All oligomers have been synthesized by SPSS. The synthesis of the core oligomers *CO* (id: **991**) and *CON* (id: **1106**) has been described in detail by *Klein et al. (2016, 2018)* and their analytical data can be found there. The synthesis of DBCO-discrete PEG(dPEG)-folic acid oligomers (termed "PEG-ligands") has also been reported in detail by *Klein et al. (2018)*, however only for PEG-ligands with PEG24 (id: **1139**) or PEG48 (id: **1140**). Here, PEG-ligands without PEG (PEG0, id: **1323**), with STOTDA (N″-succinyl-4,7,10-trioxa-1,13-tridecanediamine, named "PEG3" in this manuscript, id: **1324**) and PEG12 (id: **1325**) were synthesized analogous to the PEG-ligands with longer PEG chains. Basically, Fmoc-Glu-O-2-PhiPr was coupled to the α-amine of a Lys(Dde)-loaded resin followed by N10-(trifluoroacetyl)pteroic acid to produce functional folic acid. The trifluoroacetyl group was deprotected with 25% aqueous ammonia solution: DMF = 1:1. After standard Dde deprotection (two vol % hydrazine in DMF), the lysine's ε- amine was modified with the designated dPEG chain followed by a DBCO-acid. For PEG0, DBCO-acid is directly coupled to the lysine's ε amine. For PEG3, STOTDA's (N-Fmoc-N″-succinyl-4,7,10-trioxa-1,13-tridecanediamine) succinic acid is coupled to the lysine's ε amine and the DBCO-acid to STOTDA's terminal amine after Fmoc deprotection. Special care needs to be taken when cleaving the final product from the resin, since DBCO is sensitive to high concentrations of TFA and can be converted into unreactive side-products (*Wang et al., 2014*). Therefore, a cleavage cocktail with only 5% TFA was used (DCM:TFA:TIS = 92.2:5:2.5). Cleavage duration was 60 min. The synthesis and analysis of the lipid anchor oligomers *LA* (id: **1203**) and *LAE* (id: **1223**) is described in the Supplemental Information (1. Lipid Anchor Oligomer Synthesis, 2. Chemical Structures).

## Polyplex preparation

### Core

The amount of siRNA is the key parameter determining quantities of all other reagents in polyplex formation. For measurements and in vitro experiments, polyplexes with a final concentration of 0.025 mg/ml siRNA were produced. A nitrogen to phosphate (N/P) ratio of 12 was used to determine the amount of core oligomer *CO* (Fig. 1A) relative to the amount of siRNA. The N/P ratio sets the number of primary and secondary amines in the oligomer's structure in relation to the number of phosphates in the RNA's backbone. The azide-bearing core oligomer *CON* was handled the same way as *CO* and is described when the reference system is introduced (cf. Characterization of Core—PEG-Ligand Polyplexes)

The conventional method of polyplex preparation was done with pipettes and rapid mixing in a batch wise process. The solvent—if not noted differently—was HEPES buffer pH 7.4 with 5% glucose (HBG). This buffer was used because it does not rely on salts to be isotonic, since polyplex formation relies on charge interactions that could be hampered by ions. Here, *CO* solution ($c_{CO}$ = 0.504 mg/ml) was added quickly to a siRNA solution ($c_{siRNA}$ = 0.05 mg/ml) of equal volume and mixed by rapid pipetting, achieving a final siRNA concentration of 0.025 mg/ml. Subsequently, the formulation has been incubated for 45 min.

For automated polyplex production at a T-junction, siRNA in HBG ($c_{siRNA}$ = 0.05 mg/ml) and *CO* in HBG ($c_{CO}$ = 0.504 mg/ml) or HBG with 50% acetone were loaded into two separate syringes (one ml, Hamilton) that were connected with silicon tubes (SE-200; ProLiquid) to a T-junction (PP-T-Tüllenverbinder; ProLiquid). Each syringe was driven by a separate syringe pump (LA-120, LA-160) that run at the same speed (flowrates (FR) for each pump were 0.5, 1.0, 2.0, 5.0, and 30.0 ml/h) except for experiments with a final acetone concentration of 2.5% ($c_{siRNA}$ = 0.027 mg/ml; $FR_{siRNA}$ = 0.917, 1.833, 4.583, 9.167, 55.000 ml/h; $c_{CO}$ = 3.026 mg/ml; $FR_{CO}$ = 0.083, 0.167, 0.417, 0.833, 5.000 ml/h). The final product was collected and incubated for 45 min before use. siRNA concentration in the final formulation was 0.025 mg/ml.

For controlled core polyplex production using microfluidics, the double meander channel (DMC) in Fig. 1B was used albeit without the second meander and without both S2 inlets. siRNA in HBG ($c_{siRNA}$ = 0.033 mg/ml) was loaded into S4 and *CO* in HBG or HBG with 50% acetone ($c_{CO}$ = 3.025 mg/ml) was loaded into S3. Both syringes were driven by separate syringe pumps. FR were 100 µl/h for S3 and 900 µl/h for S4, respectively. The final product was diluted with HBG to reach $c_{siRNA}$ = 0.025 mg/ml.

### Addition of lipid anchor and lipid anchor—PEG-ligand oligomers

It was determined before that 20 mol % lipid anchor oligomer (*LA* or *LAE*) or lipid anchor—PEG-ligand oligomer in relation to $n_{CO}$ offered an optimal balance between efficacy and aggregation of the final product (data not shown).

Lipid anchor or lipid anchor—PEG-ligand oligomers were added in two different ways to core polyplexes. If the complete product is assembled in one continuous process, the DMC in Fig. 1B will be used. siRNA in HBG ($c_{siRNA}$ = 0.033 mg/ml) was loaded into S4

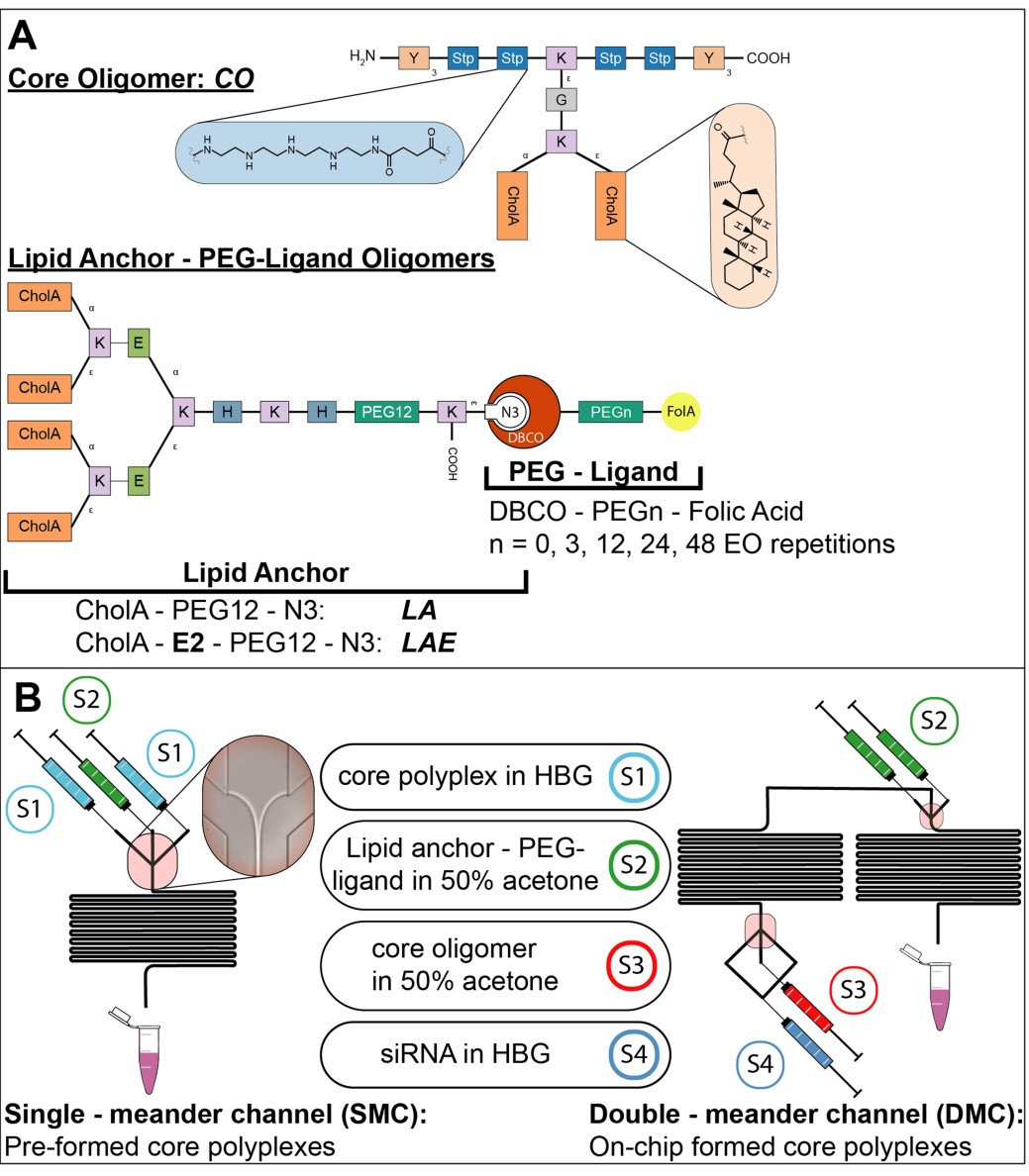

**Figure 1 Sequence-defined oligomers and their corresponding nanoparticle production method.**
(A) Oligomers used in polyplex formation: Lipid anchors were coupled to PEG-ligands before poly-
plexes were formulated. Building blocks represent natural amino acids (E = glutamic acid, G = glycine,
H = histidine, K = lysine, Y = tyrosine), synthetic building blocks (Stp = succinoyl-tetraethylene-pen-
tamine, PEG = polyethylene glycol), fatty acids (CholA = cholanic acid), and moieties for bio-orthogonal
click chemistry (N3 = azide, DBCO = dibenzocyclooctyne). (B) Production methods for polyplexes with
*CO* oligomers: Formulations used are depicted between both channels with the id of their corresponding
syringe (S1-4). Two different channels were used to produce nanoparticles during solvent exchange, a
single meander channel and a double meander channel. In the single meander channel pre-assembled
core particles were mixed with lipid anchors or lipid anchor PEG-ligand oligomers. In the double
meander channel, the complete polyplex was assembled from its starting components.

(FR = 900 μl/h) and *CO* in HBG or HBG with 50% acetone to retard siRNA compaction
($c_{CO}$ = 3.025 mg/ml) was loaded into S3 (FR = 100 μl/h). Lipid anchor or lipid anchor—
PEG-ligand oligomers in HBG with 50% acetone to facilitate solvent exchange were loaded

**Table 1 Solvents used for DLS measurements.**

| Solvent | Dispersant RI | Viscosity (cP) |
|---|---|---|
| HBG | 1.337 | 1.0366 |
| HBG (1.7% (V/V) acetone) | 1.338 | 1.0782 |
| HBG (3.3% (V/V) acetone) | 1.339 | 1.1045 |
| HBG (5.0% (V/V) acetone) | 1.340 | 1.1324 |
| HBG (6.7% (V/V) acetone) | 1.342 | 1.1750 |

Note:
    Refractive indices (RI) and viscosities in centi poise (cP).

into S2. The flow rate of each syringe S2 was 50 μl/h at a total flow rate of 1,100 μl/h, resulting in a flow rate ratio of lipid anchor oligomer to core polyplex of 1:11. The final product was diluted with HBG to $c_{siRNA}$ = 0.025 mg/ml.

Alternatively, conventionally (i.e., with pipettes) prepared core polyplexes ($c_{siRNA}$ = 0.032 mg/ml, $c_{CO}$ = 0.319 mg/ml) were fed into both inlets connected to syringe S1 (single meander channel (SMC)) with the lipid anchor oligomers filled into syringe S2. In this case, flow rates were 126.5 μl/h for S2 and 600 μl/h for each S1 resulting in a flow rate ratio of 1:10.5. The final product was diluted with HBG to $c_{siRNA}$ = 0.025 mg/ml. The difference in flow rates between the two set-ups is due to separate optimization steps. Both set-ups resulted in large volumes of core solution and only a thin stream (see Fig. 1B) of lipid anchor solution at the junction, accelerating the solvent exchange from 50% to 4.8% acetone and facilitating the association of the hydrophobic lipid anchor with the fatty acids in the core's structure. It is always indicated which method for producing core-lipid anchor-PEG-ligand polyplexes was used.

## Characterization
### DLS measurement
For dynamic light scattering (DLS) measurements, samples were prepared to contain 1.5 μg siRNA in 60 μl HEPES buffered glucose pH 7.4 (HBG) at 25 °C and the corresponding amount of oligomer. Refractive index and viscosity of the solution were calculated using the solvent builder integrated into the software (Zetasizer family software update v7.12). Viscosities and refractive indices (RI) are reported in Table 1. RI of all particles was estimated to be 1.45. In case of a *CO* core with N/P 12 and 20 mol % of *LA*, 16.6 and 2.8 μg were used, respectively. For size measurements, light scattering was measured at a 173° angle (backscatter) with a flexible attenuator with a Zetasizer Nano ZS ZEN 3600 (Malvern Panalytical Ltd, Malvern, UK) in DTS1070 micro cuvettes (Malvern Panalytical Ltd, Malvern, UK). Samples were measured three times with 12–15 sub runs each. The mean $z$-average in nm of those three runs is reported with error bars corresponding to the 95% confidence interval of the three runs. The underlying intensity distribution is depicted as violin plots in order to gain a better understanding of the formulation's size distribution. The extension of the violin plot in $x$ direction corresponds to the percentage of the total intensity measured at the specific hydrodynamic diameter depicted on the $y$-axis.

If zeta potential is measured, the sample will be taken from the cuvette after the size measurement, diluted with HBG to 800 µl and reloaded into the same cuvette. Light scattering was measured at a 90° angle with a flexible attenuator. Samples were measured three times (main runs) with enough sub runs to gather more than 10,000 total counts (usually 12–15). The mean zeta potential of those three runs is reported with error bars corresponding to the zeta deviation's mean of each main run.

### Stability of the core formulation

Core polyplex formulations were prepared using the SMC (Fig. 1B) set up as described above. *CO* was diluted in HBG with 50% acetone and siRNA was diluted in HBG only. $c_{siRNA}$ of the final solution was 0.025 mg/ml. Size, polydispersity index (PDI), and zeta potential were measured as described under "DLS Measurement." This protocol, however, was changed in the following way to allow for multiple measurements over time: Two samples with 60 µl each were prepared. The first sample was used to measure size and PDI. The second sample was diluted with HBG to 800 µl to enable zeta potential measurements. Both samples were measured directly after each other for 90 min.

### Gel shift

A 1% (w/w) suspension of agarose in Tris/Borate/EDTA (TBE) buffer (149 mM TRIS, 89 mM boric acid, two mM EDTA in demineralized water) was heated until the agarose was dissolved. After a short cooling period, 0.1% GelRed® 10000× (Biotium Inc., Fremont, CA, USA) was added. The mixture was cast into its mold and a comb was added to create wells. After 30 min, the solidified gel was placed in an electrophoresis chamber and completely immersed in TBE buffer. Polyplexes were prepared as described above. Naked siRNA was used as positive control. $C_{siRNA}$ was 0.025 mg/ml in all samples, sample volume was 20 µl. Four µl loading buffer (8.21 mM glycerol, 60 mM EDTA, 0.003 mM bromophenol blue in purified water) was added to every sample ($V_{total}$ = 24 µl) and each was pipetted in a well in the solidified gel. The gel was run for 60 min at 80 V.

For serum gel shifts, polyplexes were produced with higher siRNA concentration ($c_{siRNA}$ = 0.25 mg/ml) and diluted afterwards with FBS 1:10 to reach the desired $c_{siRNA}$ = 0.025 mg/ml. Samples containing FBS were incubated at 37 °C up to 24 h until the loading buffer was added and they were pipetted into the gel's wells. ImageJ (v. 1.52n) (*Schindelin et al., 2012*) was used to conduct a densitometry analysis of the siRNA bands. To this end, ImageJ was used to extract gray values from the respective siRNA stains. The sum of gray values as a function of the gel's extension in *y* (width of the stains) and *x* (length of the whole gel) direction was plotted with ImageJ to produce the desired analysis. The plot's arbitrary values on the *y*-axis correspond to the sum of all gray values over the full width (*y*) at a given length position (*x*). The length position x is plotted on the *x*-axis.

### FRET—experiments

Polyplexes were prepared conventionally (cf. Polyplex Preparation), albeit with a 1:2 siRNA-Cyanine 5 (Cy5):siRNA mixture. Lipid anchors (*LA* or *LAE*) were incubated with 0.75 eq. DBCO-PEG4-Atto488 (relative to azide content) over night at room temperature. Afterwards, the modified lipid anchor solution was diluted 1:2 with unmodified lipid

anchor solution, resulting in a theoretical degree of labeling of 37.5%. The lipid anchor was added to the polyplexes using the SMC (cf. Addition of Lipid Anchor and Lipid Anchor—PEG-ligand Oligomers). The final siRNA concentration was $c_{siRNA} = 0.1$ mg/ml. Therefore, the final Cy5 and Atto488 concentrations were 6.1 and 21.3 µmol/l, respectively. A total of 30 µl of each sample was filled into a 96 well plate and measured with a TEKAN pleat reader (Tecan Trading AG, Switzerland, Spark 10M, SparkControl V 2.1) with the following set of filters: Cy5: excitation wavelength: 625 nm, bandwidth 35 nm; emission wavelength: 680 nm, bandwidth 30 nm; Atto488: excitation wavelength: 485 nm, bandwidth 20 nm; emission wavelength: 535 nm, bandwidth 25 nm; Förster resonance energy transfer (FRET): excitation wavelength: 485 nm, bandwidth 20 nm; emission wavelength: 680 nm, bandwidth 30 nm. Measured fluorescence is divided by gain's value to exclude amplifier effects.

### Polyplex compaction and heparin competition assay

Core polyplexes were prepared conventionally (cf. Polyplex Preparation). Solvents were HBG and HBG with 50% acetone for core polyplexes and lipid anchor oligomers, respectively. 20 mol % of indicated lipid anchor oligomers were attached to the polyplexes via solvent exchange inside the microchannel (Fig. 1B, SMC). The final solvent was HBG with 3.3% acetone. A total of 20 µl of this mixture containing siRNA (0.025 mg/ml), *CO* (0.252 mg/ml), and lipid anchor (*LA*: 0.022, *LAE*: 0.023 mg/ml) were pipetted into a 96 well plate and incubated with 10 µl heparin solution (11.0; 55.0; 110.0; 165.0 IU/ml in HBG) or HBG for 15 min. Afterwards, 80 µl of a 0.5 µg/ml EtBr solution in HBG were added and the samples were incubated for another 5 min. When EtBr intercalates into DNA or RNA it emits a strong signal when excited. This process can be inhibited by compacting the nucleic acid with polycations. Therefore, EtBr's fluorescence correlates with the compaction efficiency of target oligomers. The addition of heparin tests the formulation's resistance against anionic stress. The fluorescence of all samples was measured with a TEKAN plate Reader (Spark 10M, SparkControl V 2.1; Tecan Trading AG, Männedorf, Switzerland) utilizing the following set of filters: Excitation wavelength: 535 nm, bandwidth 25 nm; emission wavelength: 590 nm, bandwidth 20 nm. The well containing only siRNA and EtBr served as positive control and was also used to choose optimal gain and Z-position settings. All readings were normalized to samples containing free siRNA and EtBr only (positive control) and are presented here in "(%) of positive control."

### Transmission electron microscopy

Core polyplexes were prepared conventionally or inside the SMC (cf. Polyplex Preparation). Solvents were HBG and HBG with 50% acetone for core polyplexes and lipid anchor oligomers, respectively. A total of 20 mol % of indicated lipid anchor oligomers were attached to the polyplexes using solvent exchange inside the microchannel (Fig. 1B, SMC). The final solvent was HBG with 3.3% acetone. Carbon coated copper grids (300 mesh, 3.0 mm O. D.; Ted Pella, Inc., Redding, CA, USA) were hydrophilized with a plasma cleaner under argon atmosphere (420 V, 1 min). The grid's activated surface was placed face down on a 10 µl sample droplet for 3 min. Afterwards, the sample was removed

with a filter paper and five μl staining solution (1.0% uranyl formate in purified water) was placed on the grid and immediately removed to wash the sample off. Staining was performed with the same staining solution for 5 s. Afterwards, it was siphoned off with a filter paper and the remaining liquid was left to evaporate for 20 min. Grids were stored at room temperature. Samples were measured with a JEOL JEM-1100 electron microscope at 80 kV acceleration voltage.

## Preparation of microfluidic channels
### PDMS channels
The microfluidic channels design was realized on a silica wafer with soft lithographic methods. The master microstructure was designed with the LPKF CAD/CAM software (LPKF Laser and Electronics) and made using SU8 process on silicon wafer. The microstructure of ~72 and ~90 μm thickness for single- and double-meandering channel, respectively, was rastered using LPKF ProtoLaser LDI UV-laser (LPKF Laser and Electronics). Utilized SU-8 3000 photoresist was processed in accordance with the manufacturer's instructions. The SU-8 master was subsequently silanized in an evacuated desiccator for 12 h with tri-chloro(1H,1H,2H,2H-perfluorooctyl)silane. The PDMS elastomer was mixed with 10% (w/w) crosslinker, degassed, poured onto the wafer, and cured (75 °C, 4 h). Subsequently, PDMS was peeled from the wafer, holes for the inlets were pierced at the designated positions with a biopsy puncher, and it was bonded to a glass slide by oxygen plasma-induced oxidation (Diener Electronic; 10 W high frequency generator power, 12 s, Pico Model E). The chip was left alone for 1 h to allow the reaction to complete. Afterwards, polyethylene tubes (length = 110 mm, inner diameter = 0.38 mm) were fitted into the holes in the PDMS and everything was covered with another layer of PDMS treated in the same way as mentioned above to seal the in- and outlets completely. A to-scale model of both channels' layout can be found in the Supplemental Information together with a detailed description of the channel's dimensions and calculations of Reynold's and Dean's numbers (3. Channel layout, Figs. S10 and S11).

## In vitro
### Culture
We used KB cells (cervix carcinoma, derived from HeLa cells) for all in vitro experiments. KB wild type cells were bought from DSZM (Braunschweig, Germany) and they were subsequently modified to code for a GFP-luciferase fusion mRNA by A. Cengizeroglu. The modified cell line is stably transcribing and translating the fusion mRNA to an eGFP-Luciferase fusion protein, which consists of two functional proteins, GFP and luciferase. The fusion protein's expression can be silenced by any siRNA that is complementary to the GFP-luciferase fusion mRNA. Here, we used siGFP. The construct's transfection process was described in A. Cengizeroglu's thesis (*Cengizeroglu, 2012*) and first use was demonstrated by *Dohmen et al. (2012)*. For each experiment, cells were freshly thawed from a liquid nitrogen storage tank and passaged at least four times before experiments were conducted. Cells were subcultured when 70–90% confluency was

reached. Culture conditions were 37 °C and 5% $CO_2$. KB cells were cultured in RPMI-1640 supplemented with 10% FBS and 1% penicillin/streptomycin (five ml with 100 U/ml and 100 µg/ml, respectively).

### Transfection

Cells were seeded into 96 well plates 1 day prior to transfection. All wells were pre-treated with 40 µl collagen solution per well (0.1 mg/ml, removed after 30 min, 37 °C). Afterwards, cells were seeded with 4,000 cells/well in 100 µl folate free Gibco™ RPMI 1640 (Fisher scientific, Hampton, NH, USA) supplemented with 10% FBS. The next day, the medium in all wells was replaced with 80 µl fresh medium (RMPI 1640, FolA free) and 20 µl sample solution or HBG (negative control) was added. Samples were prepared completely inside the microfluidic channel (cf. Polyplex Preparation & Fig. 1B, DMC). siRNA concentration was five µg/ml in each well. Samples were always prepared in quintuplicates. Medium was exchanged again after 4 h, total incubation time was 48 h at 37 °C, 5% $CO_2$.

### Luciferase assay

Plates were taken from the incubator and all media was removed. A total of 100 µl/well lysis buffer (Luciferase Cell Culture Lysis 5X Reagent, Promega, diluted 1:10 with purified water) was added and incubated for another 45 min at room temperature. Plates were frozen at −80 °C until measurement. A total of 35 µl/well of the cell lysate were transferred to white, opaque 96 well plates (BertholdTech, Bad Wildbad, Germany) and measured with a Centro LB 960 luminometer (BertholdTech CENTRO, Driver V. 1.21, MikroWin, V. 5.2, 10 s integration/well). A total of 100 µl LAR buffer per well (20 mM glycylglycine, 1.0 mM $MgCl_2$, 0.1 mM EDTA, 3.29 mM DTT, 0.548 mM ATP, 1.30 µM coenzyme A, adjusted to pH 8.5 with NaOH) were automatically added by the machine. The output of this measurement is relative light units (RLUs) per well. The raw data was handled the following way. The mean value from each sample was calculated and was set in relation to the mean value of the respective negative control. Results are depicted in "RLU (%) of HBG." Error bars represent 95% confidence intervals of five samples.

### MTT assay

Plates were taken from the incubator and 10 µl/well 3-(4,5-dimethylthiazol-2-yl)-2,5-diphenyltetrazolium bromide (MTT; Carl Roth, Karlsruhe, Germany, five mg/ml in PBS) were added and everything was incubated for another 2 h at 37 °C. Afterwards, the fluids were removed and the plates were frozen at −80 °C for at least 1 h. A total of 100 µl/well DMSO were added and the plates were gently shaken at 37 °C for 20 min to dissolve the purple formazan dye. The absorbance at 590 nm of each well against the reference wavelength (630 nm) was measured with a TEKAN plate reader (Spark 10M, SparkControl V 2.1; Tecan Trading AG, Männedorf, Switzerland). The raw data was handled the following way. The mean value from each sample was calculated and was set in relation to the mean value of the respective negative control. Therefore, results are depicted in "(%) of HBG." Error bars represent 95% confidence intervals of five samples.

### Dose titration

Core polyplexes were prepared conventionally with pipettes as described under "Polyplex Preparation." siRNA concentrations were chosen to have a final amount of 100, 250, 500, 750, and 1,000 ng/well. *CO* concentrations were adjusted accordingly. To be precise, siRNA concentrations in 20 µl transfection volume were (mg/ml): 0.0050, 0.0125, 0.0250 0.0375, 0.0500. *CO* concentrations were (mg/ml): 0.0458, 0.1145, 0.2291, 0.3436, 0.5041. A total of 20 µl/well of each sample was transfected as described under "Transfection." Samples were transfected in quintuplicates. The formulations' effect on luciferase activity and metabolic activity was evaluated with a luciferase assay and a MTT assay as described above.

### Data analysis

Data was analyzed with R (*R Core Team, 2018*) and RStudio (*RStudio Team, 2018*). We always report means with 95% confidence intervals, except for zeta potential measurements. Mean zeta potential was reported ± mean of zeta deviations to allow for a better understanding of the underlying zeta distribution.

Data from cell culture experiments was normalized to its negative control, which was always on the same well plate as the respective samples.

A multifactorial two—way ANOVA was used to compare mean RLU reduction of core (*CO* + siRNA) polyplex formulations with two different lipid anchor oligomers and six different PEG-ligand oligomers.

A multifactorial two—way ANOVA was used to compare mean RLU reduction of core (*CON* + siRNA) polyplex formulations with six different PEG-ligand oligomers at four different concentrations.

After each ANOVA, post hoc two-sided student's *t*-tests were conducted between all samples. Test results were corrected for the family-wise error with Holm's method. Significance was set to α < 0.05.

R code and raw data are made available here: https://doi.org/10.6084/m9.figshare.7971329.v1.

## RESULTS

The aim of our study is to demonstrate the precise production of multi-component polyplexes with a modular two-step microfluidic set-up. The device employs flow-focusing in combination with solvent exchange to allow for the successive assembly of multi-component nanoparticles. We show that the approach results in well-defined and reproducible polyplexes with controlled surface characteristics. It is used here to vary the surface layer in order to identify structure activity relationships between PEG-ligand length and transfection efficiency. Finally, we compare the findings with conventionally (educts are mixed manually with pipettes) prepared polyplexes.

### Design of the delivery systems

Oligomers for the formation of core polyplexes are designed to bind siRNA via electrostatic interactions and stabilize the resulting particle with its hydrophobic domains.

Solid phase supported synthesis (SPSS) is used to allow for precise control over the oligomers' sequence (Fig. 1A). Core oligomers (*CO*) feature four cationic Stp units that are flanked by three tyrosines (Y) on each side for aromatic and hydrophobic stabilization (*Troiber et al., 2013a*). Lysines (K) are used to introduce a branch in the main chain for the attachment of two cholanic acids (CholA) for further stabilization (*Fröhlich et al., 2012*; *Schaffert et al., 2011*) and to provide attachment points for lipid anchor oligomers. Glycine (G) is used as a spacer.

The lipid anchor oligomers (Fig. 1A, *LA, LAE*) are designed to adsorb to the core polyplexes via hydrophobic interactions between cholanic acids. In addition, they feature a histidine–lysine–histidine (H-K-H) motif to adjust solubility. The PEG12—chain exposes the terminal azide to the surrounding solution, increasing its accessibility to alkyne-bearing entities. The two glutamic acids (E) in *LAE*'s sequence increase attachment to positively charged core polyplexes and further adjust solubility. Formulation of core-lipid anchor polyplexes requires lipid anchors to be deposited on the core polyplex's hydrophobic patches during solvent exchange inside the microchannel. This step is crucial for controlling the hydrodynamic diameter of generated nanoparticles, since adding lipid anchor oligomers manually yields a suspension of polydisperse aggregates (Supplemental Information, 4. Manual formulation of core-lipid anchor polyplexes, Fig. S12).

Functional structures of interest can be coupled to lipid anchor oligomers by azide—alkyne click chemistry. This modification is possible either before deposition of lipid anchors on core polyplexes or afterwards. Here, PEG-ligand oligomers (Fig. 1A) were attached to lipid anchors 24 h before formulation with core polyplexes. The PEG-ligands have been used to investigate the influence of PEG length on transfection efficiency. They feature one dibenzozyclooctyne (DBCO) moiety, a PEG chain, and one molecule folic acid (FolA). The DBCO group enables the rapid and copper free reaction with azide groups, while the folic acid moiety facilitates binding folic acid receptors. PEG chains serve two purposes in this design: firstly, to shield the core polyplexes' positive charge, and secondly to expose folic acid to the environment. Their influence is investigated by using PEG chains of various lengths (number of EO repetitions: 0, 3, 12, 24, or 48). Lipid anchors and PEG-ligands were coupled 24 h prior to polyplex formulation. Since lipid anchors already feature a PEG12 chain, lipid anchor—PEG-ligand oligomers have a total number of 12, 15, 24, 36, or 60 EO repetitions. Polyplexes from *CO* oligomers with siRNA and lipid anchor—PEG-ligands are named after their total number of EO repetitions, for example "*P12-24F*" for polyplexes with lipid anchors with DBCO-PEG24-FolA modification.

## Polyplex characterization

Multiple experiments characterizing polyplexes can only contribute viable information about a formulation, if it is ensured that the starting formulation is in equilibrium at the time of each experiment. *Troiber et al. (2013b)* have found particles assembled from the same class of oligomers to be stable over 3 weeks. Here, we have investigated the changes in size, PDI and zeta potential of our core formulation over 90 min (Supplemental Information, 4. Core Polyplex Stability, Fig. S13). We saw no changes in size and PDI. We did note some changes in zeta potential up to 40 min, which is the reason why formulations were always used after 45 min incubation time.

### Size

Core polyplexes (*CO* + siRNA) with comparable properties were generated either by conventional bulk mixing, or at a T-junction, or with microfluidics. Integrating lipid anchor or lipid anchor PEG-ligand oligomers increased size and PDI moderately.

Hydrodynamic diameter, zeta potential, and PDI of polyplexes was measured by DLS (Figs. 2 and 3) and sizes were confirmed with transmission electron microscopy (TEM) (Fig. 4). Compacting siRNA conventionally with core oligomers (*CO*) by rapid pipetting yields particles with a mean hydrodynamic diameter ($d_Z$) of 84 nm (Fig. 2A). The PDI is very low (PDI < 0.20; Fig. 2B). Increasing control over this process either at a T-junction or with a microfluidic device, however, needs certain additional conditions to be met in order to produce similar particles. At a T-junction, the total flow rate needs to be very high (here: 60 ml/h) to generate particles with a hydrodynamic diameter of 97 nm and a PDI < 0.20. Addition of acetone does only lead to comparable particles and PDIs when flow rates of both components are identical, and *CO* is dissolved in 50% acetone as depicted in Fig. 2 ($d_Z$ = 104 nm, PDI < 0.23). This approach, however, results in an acetone concentration of 25% in the final product requiring additional efforts by evaporation or dialysis to remove the organic solvent when using it in vitro or in vivo. The complete data set (influence of flow rate, acetone and flow rate differences of 1:2 or 1:10) can be found in the Supplemental Information (6. Polyplex production at a T-junction, Fig. S14).

When preparing polyplexes, it is paramount to decrease diffusion lengths or to increase time needed for efficient siRNA compaction. Otherwise, influence on kinetically controlled polyplex formation is decreased and nanoparticles' size and polydispersity increases. Diffusion lengths inside the micro channel were minimized by flow rate differences > 1:10 and acetone was used to retard siRNA compaction. Previous experiments have shown that feeding the two outer channels with diluted siRNA solution and the middle channel with concentrated *CO* solution generates particles in an acceptable size range (data not shown). Here, a substantial difference in PDI and hydrodynamic diameter was observed when polyplexes were generated with ($d_Z$ = 95 nm, PDI < 0.14) or without ($d_Z$ = 149 nm, PDI < 0.11) additional acetone (Fig. 2).

The influence of lipid anchor and lipid anchor—PEG-ligand oligomers on core polyplexes was investigated. To this end, *LA* or *LAE* with or without their respective PEG-ligand oligomers (Fig. 1A) were attached to conventionally prepared core (*CO* + siRNA) polyplexes inside the microchannel (SMC, Fig. 1B). As described in detail in the methods section, it is essential to use concentrated lipid anchor solutions and diluted core polyplex solutions. This setting ensured that only a thin stream of lipid anchor solution is flowing through the Y-junction, accelerating the solvent exchange from 50% to 4.8% acetone and facilitating the association of the hydrophobic lipid anchor with the fatty acids in the core's structure. Polyplexes' particle size, PDI, and zeta potential were measured by DLS (Fig. 3). In order to gain a better understanding of the intensity distribution, violin plots are provided in addition to z-average values. Therefore, *z*-average values can be better assessed based on the underlying distribution, be it mono- or multimodal. The expansion
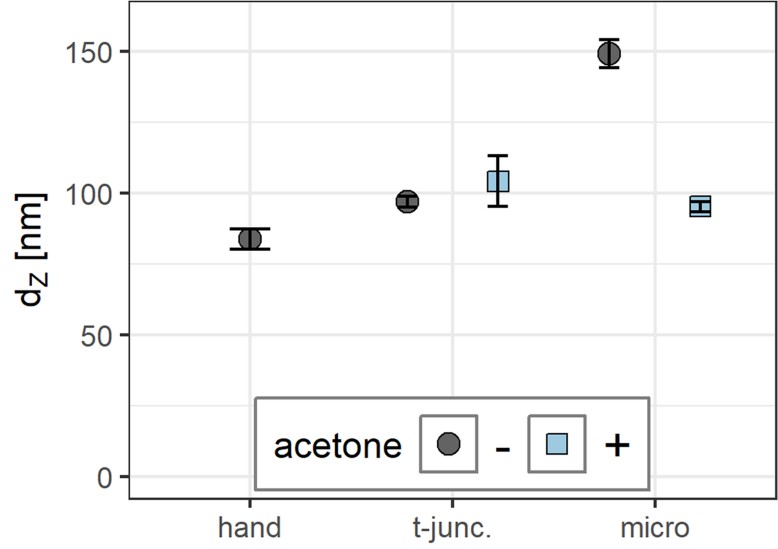

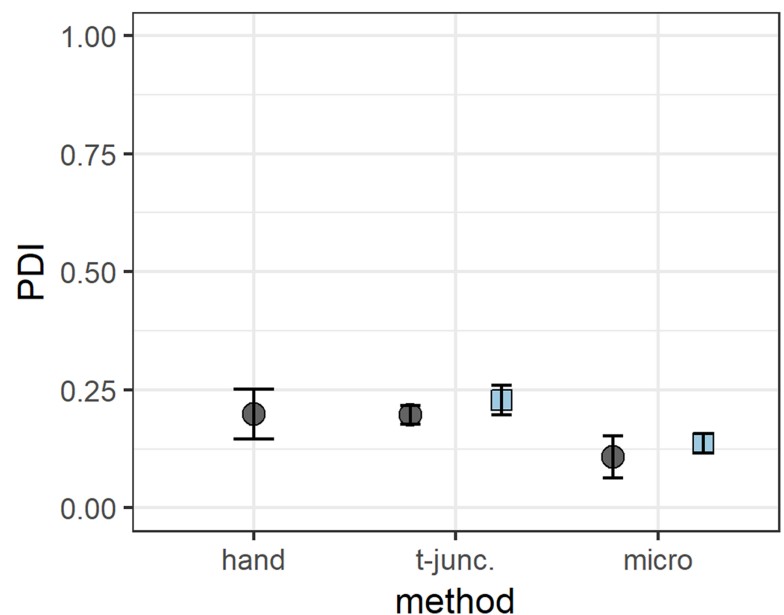

**Figure 2 Comparison of core polyplex (*CO* + siRNA) production methods.** (A) Mean hydrodynamic diameter in nm. (B) Mean polydispersity index (PDI). Method key: hand: mixing equal volumes of *CO* and siRNA solution by vigorous pipetting. T-junc.: mixing equal volumes of *CO* and siRNA solution (with or without 50% acetone) at a T-junction at 60 ml/h total flow rate. Micro: mixing an 11× larger volume of *CO* with siRNA solution (with or without 50% acetone) inside the single meander channel at 1.326 ml/h total flow rate. Grey spheres: no acetone was used. Blue cubes: acetone was used. Error bars correspond to 95% confidence intervals; $N = 3$.

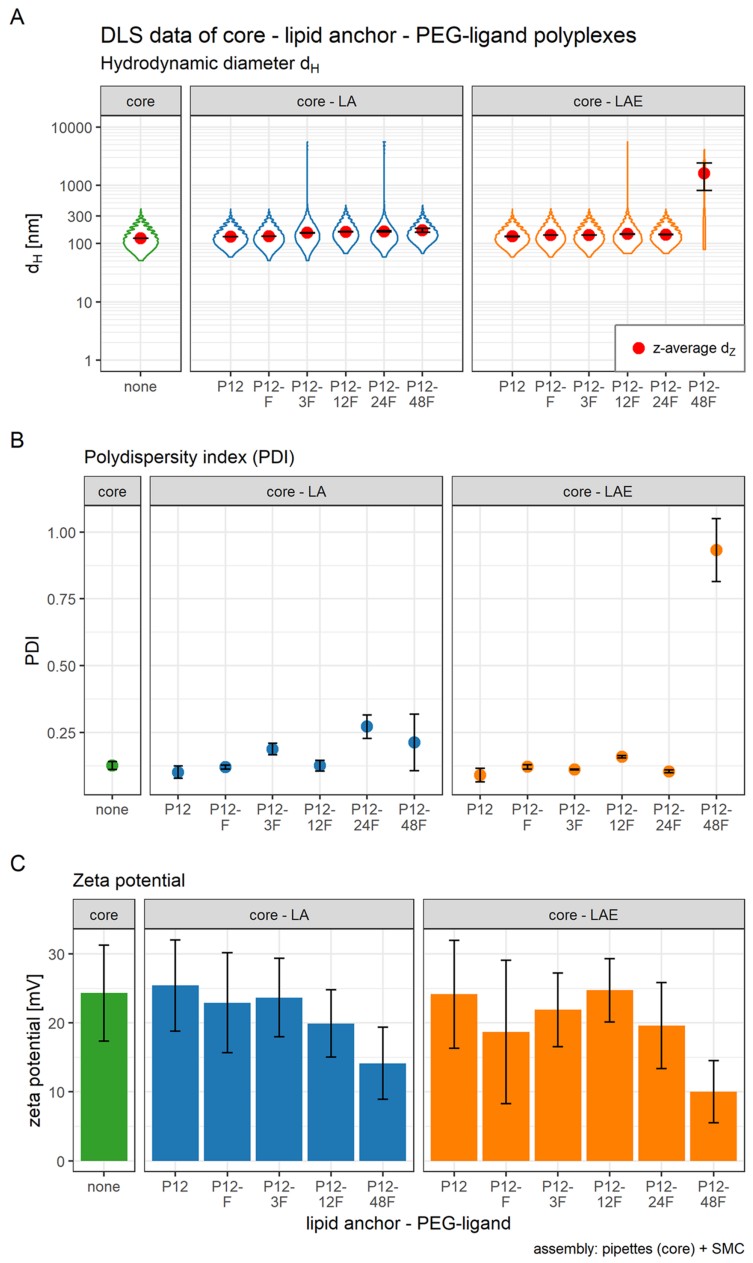

**Figure 3 Hydrodynamic diameter (d_H), PDI, and zeta potential of core, core-lipid anchor, and core-lipid anchor—PEG-ligand polyplexes.** Subfigures are divided in three panels. "core" (green) depicts particle properties of the core polyplex formulation used for all subsequent modifications with 20 mol % lipid anchor and lipid anchor-PEG-ligands. "core-*LA*" (blue) and "core-*LAE*" (orange) indicate the lipid anchor oligomer used for attaching PEG-ligands to the core polyplex. Formulation key: P12-xxF: number of ethylene oxide repetitions from lipid anchors + PEG-ligands, F: Folate. Detailed PEG-ligand description in Fig. 1A. (A) Polyplexes' hydrodynamic diameter with mean *z*-average (red dots) and respective intensity distribution depicted as violin plot (extension in *x* direction corresponds to the percentage of the total intensity measured at the specific size depicted on the *y*-axis). (B) Polydispersity index (PDI). (C) Zeta potential measured in HBG pH 7.4. Caption states assembly method: core polyplexes were prepared with pipettes, lipid anchors were added with the single meander channel (SMC). Statistics: (A) and (B) Error bars correspond to 95% confidence intervals. (C) Error bars correspond to mean zeta deviations. N = 3.

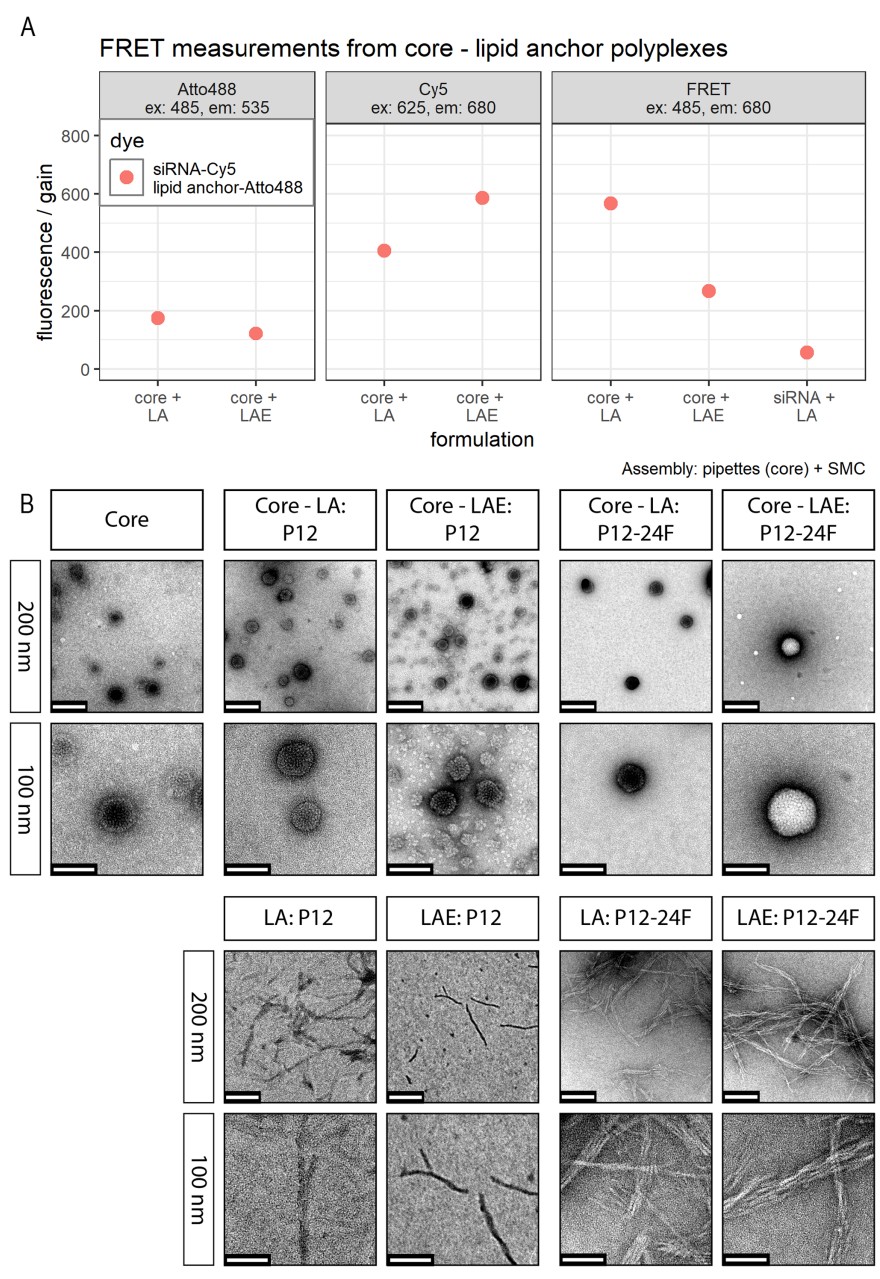

**Figure 4 FRET and TEM measurements of core (*CO* + siRNA)-lipid anchor polyplexes and their components.** (A) Title of each panel indicates dye measured. FRET: excites Atto488 (485 nm), measures Cy5 (680 nm). Color indicates dyes used in this formulation. "Sample" specifies formulation composition: "core + *LA*": core polyplex with 20 mol % *LA* oligomers. "siRNA + *LA*": control formulation without core oligomers, that is, no particle formation. Cy5 is coupled to sense strand of siRNA. Atto488 is coupled via azide-alkyne click chemistry to the azide of *LA* or *LAE* oligomers. Measured fluorescence is divided by gain's value to exclude amplifier effects. Assembly: core polyplexes were prepared with pipettes, lipid anchors were added with the single meander channel (SMC). (B) Vertical label: scale represented by white bar of respective row. Horizontal label: formulation visible in the respective column. "Core": core polyplex. "Core-*LA/LAE*: P12": core-lipid anchor polyplex. "Core + *LA/LAE*: P12-24F": core-lipid anchor—PEG-ligand polyplex. Columns without "core" depict unformulated lipid anchors or lipid anchor—PEG-ligand oligomers in solution.

in $x$ direction represents the relative frequency the respective size has been measured. The $z$-average is located close to the position with the largest expansion in $x$ direction for mono modal distributions (e.g., in the panel labeled "core"). When the distribution is multimodal, however, $z$-average's position can be quite misleading (e.g., in the panel core-*LAE*, sample *P12-48F*) and the intensity distribution needs to be considered.

The effect of adding 20 mol % (relative to $n_{CO}$) lipid anchor or lipid anchor—PEG-ligand oligomers to core polyplexes depended on the PEG-ligand's length on the respective lipid anchor. The addition of *LA* containing oligomers to the core formulation ($d_Z$ = 123 nm, PDI < 0.13) increased hydrodynamic diameters of resulting nanoparticles moderately from 131 nm (*LA* alone) to 169 nm (*LA: P12-48F*). Additionally, PDI decreased with the addition of *LA* (PDI < 0.11) or *LAE* (PDI < 0.10) oligomers and increased from PDI < 0.12 to PDI > 0.20 with longer PEG-ligands. The hydrodynamic diameter of *LAE* containing polyplexes was generally ~15 nm smaller than in *LA* containing formulations. Although, *LAE* oligomers with longer PEG-ligands were more likely to form aggregates (*LAE: P12-48F*). As expected, zeta potential of core polyplexes alone in HBG was positive with ZP = 24 mV due to the high N/P charge ratio. Incorporation of 20 mol % *LA* or *LAE* with or without PEG-ligands had only limited effect on the particles' zeta potential except for particles with *P12-48F* PEG-ligands (Fig. 3C). Incorporation of *LA:P12-48F* or *LAE: P12-48F* decreased mean zeta potential to 14 and 10 mV, respectively.

Finally, after having scrutinized all steps independently, core-lipid anchor—PEG-ligand polyplex production from its single components inside one microchannel was investigated. The DMC (Fig. 1B) was used. Syringe S3 was filled with siRNA in HBG and S4 with *CO* in HBG with or without 50% acetone. Syringes S2 were loaded with four different oligomers in HBG with 50% acetone: *LA, LA: P12-24F, LAE,* or *LAE: P12-12F*. Eight runs were conducted, each lipid anchor or lipid anchor—PEG-ligand oligomer was mixed with core polyplexes produced with or without the aid of acetone. Sizes were comparable to core-lipid anchor—PEG-ligand polyplexes with conventionally prepared cores when no acetone was used in the core production step. When *CO* was dissolved in 50% acetone, however, polyplexes completely prepared with microfluidics had a smaller hydrodynamic diameter and PDI (Supplemental Information, 7. Controlled Production of Core-Lipid Anchor—PEG-Ligand Polyplexes from their Single Components, Fig. S15).

### Lipid anchor integration

*LA* and *LAE* integrate into core polyplexes.

Investigation of *LA* and *LAE* integration into core polyplexes was carried out by TEM and FRET measurements. TEM measurements revealed that formulations from siRNA and *CO* form spherical particles with a diameter <100 nm (Fig. 4B) which is in good agreement with DLS measurements (Fig. 3A). There was no obvious difference when particles were produced conventionally or with microfluidics (Supplemental Information, 8. TEM: Comparisons of polyplexes produced with pipettes or with the DMC, Fig. S16). *LA* and *LAE* with or without covalently bound PEG-ligands alone form tubular or fibrous structures on the TEM grid that could not be found when formulated together with core polyplexes. This finding suggests that lipid anchor oligomers are indeed interacting with core polyplexes.

These findings obtained by TEM were supported by FRET measurements. Receiving measurable FRET signals implies a distance <10 nm between chromophores (*Clegg, 1995*; *Förster, 1948*). Here, 50% siRNA with one molecule Cy5 on the sense strand was used for conventional core polyplex formation. Lipid anchor oligomers were modified with 0.75 equivalents (relative to lipid anchors' azide) DBCO-PEG4-Atto488 and subsequently deposited on the conventionally prepared core polyplex using solvent exchange inside the micro-channel (SMC). These polyplexes emitted strong FRET signals when Atto488 dyes were excited and fluorescence was measured from Cy5 dyes alone (Fig. 4A). When *CO* was missing from the formulation, polyplex formation did not occur making energy transfer between dyes a function of their dilution only (sample "siRNA + *LA*" in panel "FRET" in Fig. 4A). All control experiments (FRET measurements from polyplexes with only one dye and fluorescence measurements of both dyes separately) can be found in the Supplemental Information (9. FRET control experiments, Fig. S17).

### Stability

Lipid anchors do not influence stability of core polyplexes.

Polyplex stability was assessed with two different methods. The general ability of polyplexes to compact and hold siRNA back under the influence of an electric field was investigated with an agarose gel shift assay and its densitometry analysis to simplify comparison of bands. Ability to compact siRNA and resist polyanionic stress was tested with an EtBr displacement assay with or without additional heparin.

Polyplexes from *CO* and siRNA were prepared conventionally in HBG and lipid anchors ± PEG-ligands were attached inside the micro channel (SMC). Samples were diluted 1:10 with HBG or serum (FBS). Additionally, samples containing serum were incubated at 37 °C for up to 24 h to assess stability under the influence of body temperature and serum components. There was no visible difference between all formulations at $t = 0$ h with or without additional FBS. At the 4 h mark, only small differences between samples were visible, while all samples retained most of their payload. After 24 h, core—lipid anchor or core-lipid anchor—PEG-ligand formulations revealed a slight decrease in siRNA retention capability in comparison to the core formulation alone. At this time, the core-*LAE* formulation seemed to be better at retaining siRNA than the core-*LA* formulation. When lipid anchors coupled with PEG-ligands were used, however, core-*LA*—PEG-ligand formulations retained siRNA better than their *LAE* containing counterparts (Supplemental Information, 9. Gel shift assay, Figs. S18 and S19).

Polyplexes (*CO* + siRNA) were prepared conventionally and lipid anchors were added inside the SMC for the EtBr displacement assay with and without heparin competition. In this assay, *LA* and *LAE* containing polyplexes showed an unaltered protection against dye displacement behavior. Fluorescence without additional heparin for core formulation, core-*LA* formulation and core-*LAE* formulation was 14%, 18%, and 11% of the positive control, respectively. One IU/ml heparin increased fluorescence to 37%, 39%, and 22%. Total displacement was observed at heparin concentrations above five IU/ml (Supplemental Information, 9. EtBr displacement assay, Fig. S20).

*Toxicity*

Core (*CO* + siRNA)-lipid anchor—PEG-ligand polyplexes do not alter the metabolic activity profile of KB cells in comparison to core polyplexes alone.

Different fatty acids in oligo-amidoamines have been shown to induce membrane leakage in erythrocytes and to increase cell death in in vitro cell assays (*Klein et al., 2016*; *Reinhard, Zhang & Wagner, 2017*). Influence of target formulations completely prepared with microfluidics on metabolic activity of KB cells was assessed by MTT assay to account for any apparent effects on cell survivability. The MTT assay correlates metabolic activity to the amount of formazan dye produced by oxidoreductase enzymes while consuming NAD(P)H. All formulations tested in this assay showed no reduction of formazan absorption relative to untreated KB cells (Supplemental Information: 11. MTT assay of core-lipid anchor—ligand polyplexes, Fig. S21).

## Transfection of core-lipid anchor—PEG-ligand nanoparticles

Core (*CO* + siRNA)-lipid anchor—PEG-ligand nanoparticles with *LA: P12-24F* or *LAE: P12-12F* showed the largest effect on luciferase reporter gene silencing activity in KB cells.

KB cells possessing an eGFP-luciferase fusion gene controlled by a constitutively active promoter were used in all cell experiments. Gene expression can be modulated by RNA interference: if a siRNA (here: siGFP) that is complementary to any part of the target mRNA (here: eGFP-luciferase fusion mRNA) reaches the cytosol and is incorporated into the RISC complex the corresponding mRNA will be degraded selectively. In this case, the eGFP-luciferase fusion protein expression is reduced which in turn leads to a decrease in GFP and luciferase enzymatic activity. Using an in vitro bioluminescence assay, gene silencing efficacy of the siRNA formulation can be correlated to the reduction of, in our case, luciferase activity as measured in RLUs as shown in Fig. 5. Non-siRNA dependent effects on luciferase activity were monitored with cells treated with identical polyplexes containing control siRNA only. Polyplexes were prepared from their starting materials using microfluidics (Fig. 1B, DMC). Amount of siRNA/well was optimized and set to 500 ng/well (Supplemental Information: 13. Dose Titration, Fig. S22).

Effects of lipid anchor and PEG-ligands on luciferase activity were estimated using a multifactorial two-way ANOVA. All calculated effects were statistically significant. Main effect of lipid anchors: $F(1, 48) = 8.91$, $p = 0.032$, $\omega^2 = .02$, main effect of PEG-ligands: $F(5, 48) = 14.78$, $p < 0.001$, $\omega^2 = 0.43$, and the interaction effect between PEG-ligands and lipid anchors: $F(5, 48) = 17.02$, $p < 0.001$, $\omega^2 = 0.32$.

After it was established that including lipid anchors and PEG-ligands influenced luciferase enzyme activity, post hoc student's *t*-tests (HOLM corrected) were conducted to identify the statistical significance of each comparison (Table 2). Samples with *LA* are shown in Fig. 5A, samples with *LAE* in Fig. 5B. Figure S23 compares siGFP containing samples from Figs. 5A and 5B against each other to gauge the lipid anchor's influence on the polyplexes gene silencing efficacy.

Both sets showed an effect on eGFP-luciferase gene silencing activity that is dependent on the PEG-ligand's length. For *LA* containing formulations, RLUs decreased with increasing PEG length, reached their base with *P12-24F* and rose again with *P12-48F*.

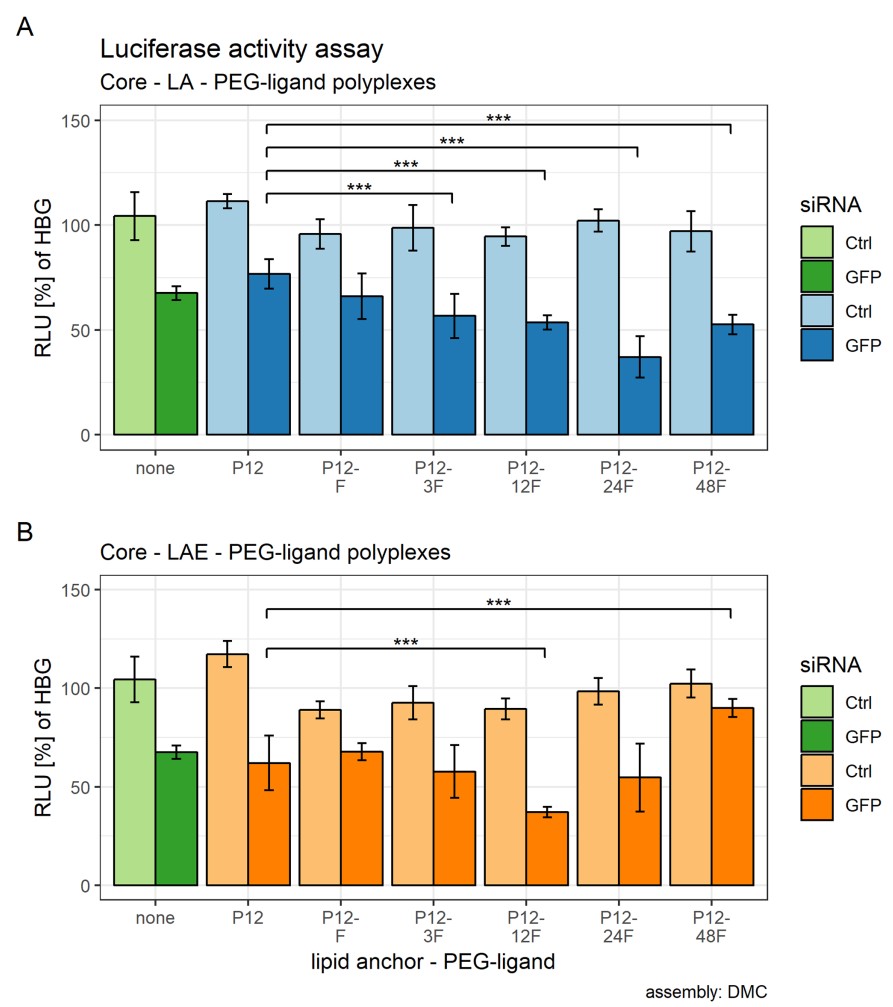

**Figure 5 Luciferase activity assay of core (*CO* + siRNA)-lipid anchor—PEG-ligand polyplexes.** Luciferase enzyme activity was measured in relative light units (RLU) and is shown relative to values of buffer treated cells. Colors indicate type of siRNA used: light color: control siRNA, saturated color: siGFPLuc siRNA. Lipid anchor—PEG-ligand key: "none" (green bars): core polyplex formulation alone; used for all subsequent modifications with 20 mol % lipid anchors and lipid anchor—PEG-ligands. P12: core polyplex with unmodified lipid anchor. P12-xxF: PEG12 from the lipid anchor + PEGxx from the PEG-ligand, F: Folate. Detailed PEG-ligand description in Fig. 1A. (A) Polyplexes with *LA* (blue bars). (B) Polyplexes with *LAE* (orange bars). Assembly: completely inside the double meander channel (DMC). Statistics: tips of horizontal lines indicate compared samples. Comparison: two-sided student's *t*-test with HOLM correction. $N = 5$. Key: NS, not significant at $\alpha = 0.05$; ***: $\alpha < 0.001$. Error bars correspond to 95% confidence intervals.

The same pattern was observed with *LAE* containing formulations, except that the base was already reached with *P12-12F* and effects of polyplexes with *P12-48F* are comparable to the siCtrl containing particles.

## Characterization of *CON*—PEG-ligand polyplexes

*CON* oligomers, in contrast to *CO* oligomers, feature an additional azidolysine N-terminally (Fig. 6A). Consequently, PEG-ligands can be coupled covalently to *CON* containing core polyplexes. Generally, azide-bearing core oligomers were modified with

**Table 2 Results of post hoc tests of core (*CO* + siRNA)-lipid anchor—PEG-ligand polyplexes.**

| Sample | *p*-value | *p*-value (corrected: HOLM) | Cohen's d |
|---|---|---|---|
| Comparison: sample to core-LA formulation | | | |
| Core-LA: P12-F | 0.018 | 0.142 | 1.45 |
| **Core-LA: P12-3F** | **<0.001** | **<0.001** | **2.79** |
| **Core-LA: P12-12F** | **<0.001** | **<0.001** | **5.21** |
| **Core-LA: P12-24F** | **<0.001** | **<0.001** | **5.72** |
| **Core-LA: P12-48F** | **<0.001** | **<0.001** | **5.00** |
| Comparison: sample to core-LAE formulation | | | |
| Core-LAE: P12-F | 0.316 | 1.000 | 0.69 |
| Core-LAE: P12-3F | 0.452 | 1.000 | 0.39 |
| **Core-LAE: P12-12F** | **<0.001** | **<0.001** | **3.09** |
| Core-LAE: P12-24F | 0.199 | 0.996 | 0.58 |
| **Core-LAE: P12-48F** | **<0.001** | **<0.001** | **3.36** |
| Comparison: core-LA—PEG-ligand vs. core-LAE—PEG-ligand | | | |
| Core-LA | 0.030 | 0.213 | 1.66 |
| Core-LA: P12-F | 0.702 | 1.000 | 0.25 |
| Core-LA: P12-3F | 0.861 | 1.000 | 0.12 |
| **Core-LA: P12-12F** | **<0.001** | **<0.001** | **6.72** |
| Core-LA: P12-24F | 0.040 | 0.237 | 1.55 |
| **Core-LA: P12-48F** | **<0.001** | **<0.001** | **10.1** |

Note:
Two-sided student's *t*-test with and without HOLM correction. Cohen's d: effect size. Magnitude: <0.2: negligible. <0.5: small. <0.8: medium. <1.20: large. >1.20: very large. Bold values are significant at $\alpha < 0.05$.
Core: core polyplex with siGFP and *CO*; *LA/LAE*: lipid anchor oligomers; Core-LA: no PEG-ligand: P12-xxF, number of PEGx from lipid anchor + PEG-ligand; F: folic acid.

PEG-ligands 45 min after polyplex formation (Fig. 6B) because coupling PEG-ligands to core oligomers before polyplex formation hampers siRNA compaction (*Morys et al., 2017*). This method has already been established by *Klein et al. (2018)* and was used here to validate results generated with core polyplexes that had PEG-ligands attached by lipid anchors.

Increasing PEG-ligand length and molar amounts promotes aggregation.

We covalently bound PEG-ligands to *CON* core polyplexes prepared as described in *Klein et al. (2018)* and depicted here in Figs. 1A and 6A. In brief, *CON* oligomers and siRNA were mixed manually and incubated for 45 min. Afterwards, PEG-ligands were added, and the azide-alkyne click reaction was allowed to complete for 4 h. Results from these covalently modified polyplexes were used to confirm results generated with the lipid anchor containing system. The main difference between both formulations is the mode of incorporation of target PEG-ligands. On the one hand, *CO* based core polyplexes need lipid anchor oligomers for the non-covalent attachment of PEG-ligands. PEG-ligands are coupled covalently to lipid anchor oligomers before the polyplex formulation process. On the other hand, *CON* based core polyplexes feature azides that enable the PEG-ligand's covalent integration into core polyplexes after core polyplex formulation.

Additionally, we increased PEG-ligand concentrations to investigate their influence on particle size as well. We found that core polyplexes modified with 25 mol % PEG-ligand

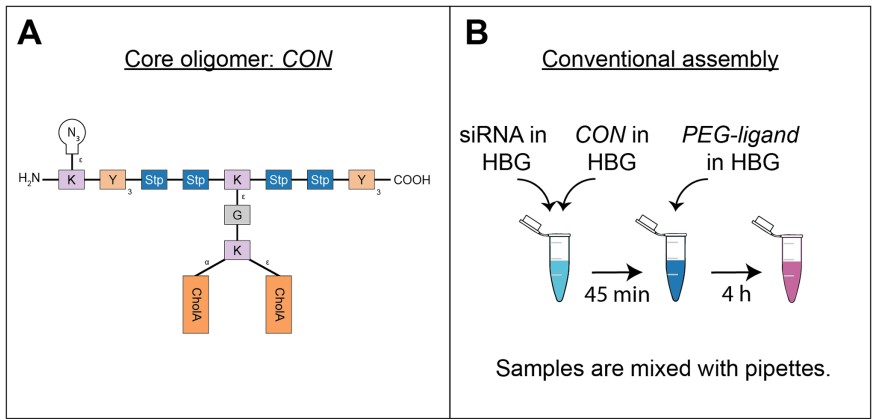

**Figure 6 Sequence-defined oligomers and their corresponding nanoparticle production methods.**
(A) Core oligomer featuring an azide (*CON*): PEG-ligands were coupled to *CON* after polyplex formation. Building blocks represent natural amino acids (E = glutamic acid, G = glycine, H = histidine, K = lysine, Y = tyrosine), synthetic building blocks (Stp = succinoyl-tetraethylene-pentamine, PEG = polyethylene glycol), fatty acids (CholA = cholanic acid), and moieties for bio-orthogonal click chemistry (N3 = azide, DBCO = dibenzocyclooctyne). (B) Manual production method for *CON*—PEG-ligand polyplexes.

were all in the same size range ($d_H \sim$ 120 nm, Fig. 7A) and PDI (~0.15, Fig. 7B), except for formulations with *P48F* ($d_H$ = 136 nm, PDI = 0.20). These results were comparable to *CO* based core—lipid anchor polyplexes with 20 mol % PEG-ligands (Figs. 3A and 3B), except with *LAE* which showed a substantial increase in size and PDI with *P48F*. Increasing PEG-ligand concentration up to 100 mol % did not substantially alter size and PDI of polyplexes with *F* ($d_H$ = 122 nm, PDI = 0.16), *P3F* ($d_H$ = 115 nm, PDI = 0.14), and *P12F* ($d_H$ = 135 nm, PDI = 0.11), but had a large effect on size and PDI of *P24F* ($d_H$ = 1,817 nm, PDI = 0.62) and *P48F* ($d_H$ = 8,393 nm, PDI = 0.67) containing particles which basically showed aggregation when functionalized with more than 25 mol % PEG-ligands.

## Transfection of *CON*—PEG-ligand polyplexes
The optimal PEG-ligand length is PEG12 or PEG24.

The influence of molar amount and PEG length of PEG-ligands on luciferase activity was estimated using a multifactorial two-way ANOVA. siCtrl polyplexes were included in addition to siGFP polyplexes to detect apparent toxicity and to attribute it to either PEG length, molar amount or both. Significant terms suggest an influence of the tested variable (PEG length and molar amount) on transfection efficiency. A significant interaction term indicates that both variables influence each other. Main effect of PEG length for siGFP: $F_{(4, 90)}$ = 3.71, $p < 0.008$, $\omega^2$ = 0.32, main effect of molar amount used with siGFP: $F_{(1, 90)}$ = 24.96, $p < 0.001$, $\omega^2$ = 0.36, interaction effect between PEG length and molar amounts with siGFP: $F_{(4, 90)}$ = 4.15, $p$ = 0.004, $\omega^2$ = 0.04.

The ANOVA with siCtrl polyplexes yielded the following results: Main effect of PEG length for siCtrl: $F_{(4, 90)}$ = 4.37, $p < 0.003$, $\omega^2$ = 0.23, main effect of molar amount used with siCtrl: $F_{(1, 90)}$ = 2.48, $p$ = 0.119, $\omega^2$ = 0.20, and the interaction effect of PEG length with molar amount with siCtrl: $F_{(4, 90)}$ = 13.52, $p < 0.001$, $\omega^2$ = 0.19.

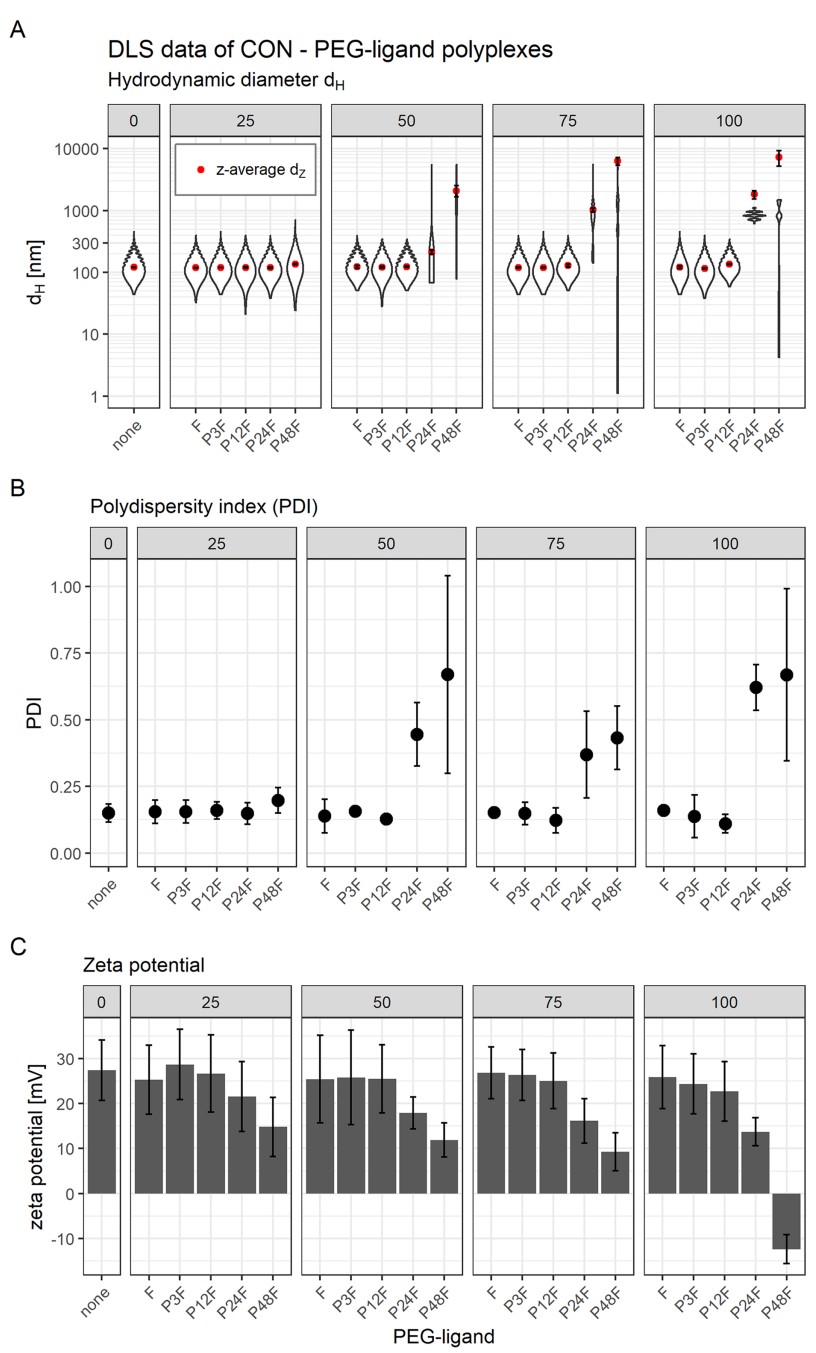

**Figure 7 Hydrodynamic diameter ($d_H$), PDI, and zeta potential of core ($CON$ + siRNA) and core—PEG-ligand polyplexes.** Subfigures are divided into five panels. Numbers indicate the amount of PEG-ligand used in mol % relative to $n_{CON}$. Formulation key: "core polyplex": unmodified $CON$—siGFP polyplex. Px: PEGx, F: folate. Detailed oligomer description in Fig. 1A (PEG-ligands) and Fig. 6A ($CON$). Assembly: conventionally with pipettes. (A) Hydrodynamic diameter ($d_H$) and mean $z$-average (red dots) with respective intensity distribution depicted as violin plot (extension in $x$ direction corresponds to the percentage of the total intensity measured at the specific size depicted on the $y$-axis). (B) Polydispersity index (PDI). (C) Zeta potential measured in HBG pH 7.4. Statistics: (A) and (B) Error bars correspond to 95% confidence intervals. (C) Error bars correspond to mean zeta deviations. $N = 3$.

**Table 3 Results of post hoc tests between core (siGFP + *CON*) polyplex formulations with and without PEG-ligands.**

| Mol % | PEG-ligand (siGFP + CON polyplex) | *p*-value | *p*-value (corrected: HOLM) | Cohen's d |
|---|---|---|---|---|
| 25 | **F** | **<0.001** | **<0.001** | **2.39** |
| | **P3F** | **<0.001** | **<0.001** | **3.08** |
| | **P12F** | **<0.001** | **<0.001** | **3.80** |
| | **P24F** | **<0.001** | **<0.001** | **2.88** |
| | **P48F** | **0.006** | **0.006** | **1.24** |
| 50 | **F** | **<0.001** | **<0.001** | **1.37** |
| | **P3F** | **<0.001** | **<0.001** | **3.74** |
| | **P12F** | **<0.001** | **<0.001** | **4.44** |
| | **P24F** | **<0.001** | **<0.001** | **1.80** |
| | **P48F** | **<0.001** | **<0.001** | **2.80** |
| 75 | **F** | **<0.001** | **<0.001** | **4.04** |
| | **P3F** | **<0.001** | **<0.001** | **5.75** |
| | **P12F** | **<0.001** | **<0.001** | **7.01** |
| | **P24F** | **<0.001** | **<0.001** | **7.15** |
| | **P48F** | **<0.001** | **<0.001** | **2.33** |
| 100 | **F** | **<0.001** | **<0.001** | **5.83** |
| | **P3F** | **<0.001** | **<0.001** | **6.84** |
| | **P12F** | **<0.001** | **<0.001** | **7.42** |
| | **P24F** | **<0.001** | **<0.001** | **7.62** |
| | **P48F** | **<0.001** | **<0.001** | **2.84** |

**Note:**
Two-sided student's *t*-test with and without HOLM correction. Cohen's d: effect size. Magnitude: <0.2: negligible. <0.5: small. <0.8: medium. <1.20: large. >1.20: very large. Bold values are significant at $\alpha < 0.05$. PEG-ligands are covalently bound to core polyplexes with siGFP and *CON*.
Mol %: $n_{PEG-ligand}$: $n_{CON}$; Px: PEGx; F: Folic acid.

Post hoc tests were used to quantify the influence of separate PEG-ligands on luciferase knockdown in comparison to the core polyplex formulation (Tables 3 and 4). Cells treated with conventionally prepared *CON* polyplexes with siGFP showed a non-significant decrease in RLUs compared to polyplexes with siCtrl. Incubating polyplexes for 4 h with targeting PEG-ligands of various lengths decreased luciferase activity significantly compared to core formulation without PEG-ligands (Table 3). Increasing PEG-ligand concentration up to 100 mol % (relative to $n_{CON}$) increased siGFP's effect as well, but toxicity and aggregation tendency increased simultaneously (Figs. 7 and 8B).

There was, however, a "sweet spot" for the positive influence of PEG-ligand length and molar amount. Increasing the number of PEG repetitions per PEG-ligand decreased RLUs up to *P12F* when 25 mol % PEG-ligand was added. Longer PEG-ligands were not as powerful (Fig. 8A, panel 25 mol %). Gradually increasing total PEG-ligand amount relative to free azides increased efficacy but lead to aggregation (Fig. 7) with associated toxicity (Fig. 8B, *P24F*) and loss of function (Fig. 8A, *P48F*) for some polyplexes with >50 mol % PEG-ligands as well. PEG-ligands with less than 24 PEG units did not exhibit aggregation or toxicity independent from the amount used.

**Table 4 Results of post hoc tests between core (siCtrl + *CON*) polyplex formulations with and without PEG-ligands.**

| Mol % | PEG-ligand (siCtrl + CON polyplex) | *p*-value | *p*-value (corrected: HOLM) | Cohen's d |
|-------|-----------|---------|---------------------|-----------|
| 25 | **F** | **<0.001** | **0.007** | **2.33** |
| | **P3F** | **<0.001** | **<0.001** | **3.18** |
| | **P12F** | **<0.001** | **<0.001** | **2.30** |
| | **P24F** | **<0.001** | **<0.001** | **3.02** |
| | **P48F** | **0.006** | **0.039** | **2.08** |
| 50 | F | 0.433 | 1.000 | 0.42 |
| | P3F | 0.627 | 1.000 | 0.36 |
| | P12F | 0.329 | 1.000 | 0.57 |
| | P24F | 0.686 | 1.000 | 0.22 |
| | P48F | 0.064 | 0.319 | 0.93 |
| 75 | **F** | **<0.001** | **<0.001** | **2.97** |
| | **P3F** | **<0.001** | **<0.001** | **4.99** |
| | **P12F** | **<0.001** | **<0.001** | **3.49** |
| | **P24F** | **<0.001** | **<0.001** | **6.10** |
| | P48F | 0.012 | 0.074 | 1.39 |
| 100 | **F** | **<0.001** | **<0.001** | **3.52** |
| | **P3F** | **<0.001** | **<0.001** | **4.75** |
| | **P12F** | **<0.001** | **<0.001** | **4.82** |
| | **P24F** | **<0.001** | **<0.001** | **15.01** |
| | **P48F** | **<0.001** | **<0.001** | **3.40** |

**Note:**
Two-sided student's *t*-test with and without HOLM correction. Cohen's d: effect size. Magnitude: <0.2: negligible. <0.5: small. <0.8: medium. <1.20: large. >1.20: very large. Bold values are significant at $\alpha < 0.05$. PEG-ligands are covalently bound to core polyplexes with siCtrl and *CON*.
Mol %: $n_{PEG-ligand} : n_{CON}$; Px: PEGx; F: Folic acid.

Producing *CON*—PEG-ligand polyplexes completely inside the DMC yielded similar results to conventionally produced polyplexes. The results from the luciferase activity assay and the MTT assay from *CON* polyplexes with 75 mol % PEG ligands are presented in the Supplemental Information (14. Luciferase and metabolic activity assay of *CON* polyplexes with PEG-ligands produced with the DMC, Fig. S24).

## DISCUSSION

We have shown that controlled production of simple two component polyplexes is feasible and that it can be extended to generate more sophisticated products. It depends on the aim of the experiment which method is most suitable. Conventional bulk mixing with pipettes is best chosen when polyplexes must be prepared quickly and high control over mixing parameters is not an issue. It is problematic, however, if bulk mixing is the default method for preparing polyplexes, since size and polydispersity are heavily dependent on the concentration of its components and their respective volumes. T-junctions are best for continuously preparing larger volumes of polyplex solutions with some control over mixing speed. Since the mixing is turbulent and flows are fast, mixing vastly different

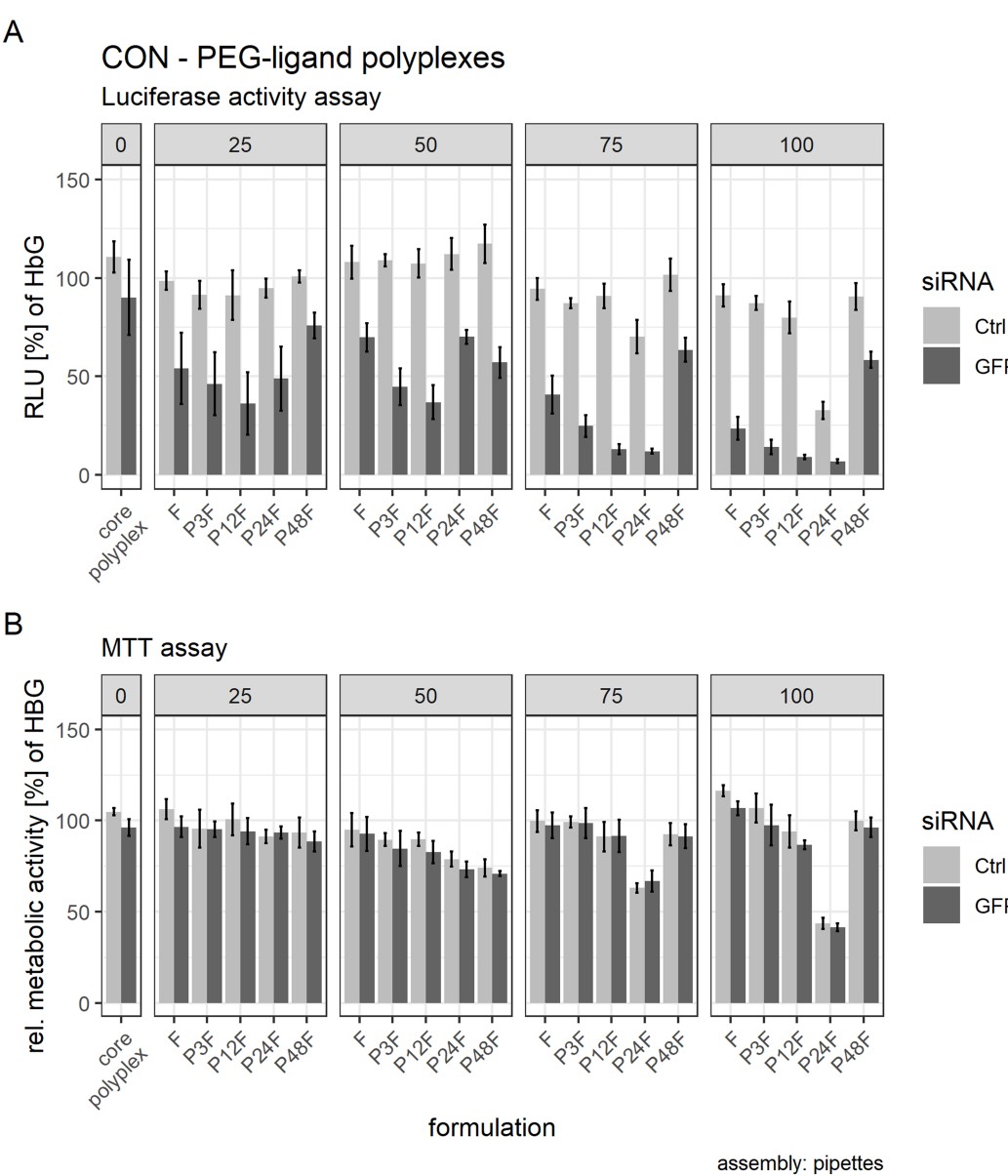

**Figure 8 Luciferase activity assay and MTT assay of core (*CON* + siRNA)—PEG-ligand polyplexes.** Polyplexes were prepared conventionally. Colors indicate type of siRNA used: light color: control siRNA, saturated color: siGFPLuc siRNA. "core polyplex" depicts particle properties of the naked core polyplex formulation used for all subsequent modifications. Panel's key: *x* mol % PEG-ligands relative to n_{CON}. Formulation key: Px: PEGx, F: Folate. Detailed oligomer description in Fig. 1A (PEG-ligands) and Fig. 6A (*CON*). Assembly: conventionally with pipettes. (A) Luciferase assay. Luciferase enzyme activity is measured in relative light units (RLU) and shown relative to values of buffer treated cells. (B) MTT assay. Values are shown relative to values of buffer treated cells. Statistics: error bars correspond to 95% confidence intervals. *N* = 5.

volumes can be challenging. Moreover, the high mixing speed required would limit the further automated processing of prepared polyplexes, if the next step involved pressure sensitive components. Microfluidics excel in producing polyplexes with a high degree of control over external mixing parameters and additional reactants, which is reflected in polydispersity indices around 0.1 for these polyplexes. Increasing throughput to

T-junction levels, however, would entail parallelization of the whole set-up, which is only feasible when proto typing and sample preparation can be automated.

To demonstrate the advantages of this approach, we have produced multi-component polyplexes from their single components in one continuous experiment which would have been impossible with bulk mixing or at a T-junction.

Morphology of core—lipid anchor polyplexes was shown with TEM and FRET experiments. TEM pictures revealed fibrous structures for samples containing only *LA* or *LAE* alone, with or without PEG-ligands. These structures, however, were not visible when core-lipid anchor polyplexes were examined. Moreover, FRET experiments showed a strong signal for labeled core (*CO* + siRNA-Cy5)-lipid anchor (Atto488) polyplexes that could not be observed in mixtures containing only siRNA-Cy5 and lipid anchor-Atto488 without *CO*. Taken together, both results indicate successful integration of lipid anchor oligomers into core structures.

The investigation of size, PDI and zeta potential of core-lipid anchor and core-lipid anchor—PEG-ligand polyplexes showed matching results. Mean hydrodynamic diameter ($d_H$) and mean PDI increased with increasing PEG-ligand length while zeta potential was gradually reduced. Zeta potential reduction could also be one reason for particles with longer PEG chains forming aggregates, since electrostatic repulsion was diminished. Similarly, polyplexes that had their PEG-ligands directly coupled to *CON* showed an increase in PDI and $d_H$ with PEG24 and PEG48 containing PEG-ligands, specifically with PEG-ligand content >25 mol %.

There is evidence that integration of PEG chains into electrostatically formed nanoparticles decreases its stability (*Morys et al., 2017*). On the one hand, this could be a critical problem if the polyplex disintegrates before it delivers its payload. On the other hand, it has been shown that increased stability has the potential to inhibit delivery as well, if the polyplex does not release its payload once inside the target cell (*Leong & Grigsby, 2010*; *Schaffer et al., 2000*). Therefore, a balance needs to be found between both extremes. Here, core-lipid anchor polyplexes and core polyplexes alone produced similar results, both when treated with poly-anions in an EtBr displacement assay with up to five IU/ml heparin and when siRNA compaction and retention is tested with a gel shift assay with or without incubation in 90% serum at 37 °C. These findings suggest an unaltered stability profile of core-lipid anchor particles compared to its naked core polyplex formulation. Even the addition of PEG-ligands to core-lipid anchor polyplexes did not alter serum gel shift results exceedingly until the formulation had been incubated for 24 h at 37 °C.

Biological activity of core-lipid anchor—PEG-ligand particles was investigated by silencing luciferase protein expression in KB cells in vitro. From previous studies, we anticipated that changing the PEG-ligand on the polyplexes would have the biggest impact on luciferase activity. ANOVA's results confirmed our hypothesis. Additionally, the results revealed a barely significant influence of the lipid anchors used. The small effect can be explained by the lipid anchor's function: since lipid anchors are designed to facilitate association with the core polyplex only, their effect pales in comparison to PEG-ligands which are especially designed to enhance uptake. However, lipid anchors apparently influence the effect of PEG length in PEG-ligands by shifting the most efficient spacer from

PEG12 for *LA (LA: P12-12F)* to PEG24 for *LAE (LAE: P12-24F)* containing polyplexes. Additionally, tendency for aggregation seems to be increased with *LAE*. One could speculate, whether the small, non-significant reduction in mean zeta potential serves and suffices as trigger for aggregation. Nevertheless, the additional E's in LAE's structure make the compound's purification easier which might be the decisive argument for the E's integration. The biological activity of polyplexes containing 20 mol % lipid anchor-PEG-ligands is comparable to polyplexes from *CON* + siRNA with 25 or 50 mol % covalently bound PEG-ligands without lipid anchors.

The predictive value of the lipid anchor containing systems has been assessed with the system published by *Klein et al. (2018)*. Here, 25 mol % PEG-ligands were covalently coupled to conventionally prepared polyplexes from *CON* and siRNA. Subsequently, KB cells were transfected. Indeed, the silencing pattern visible with core-lipid anchor—polyplexes was reproduced and the hinted-on problems with longer PEG chains—aggregations, toxicity—were also visible when PEG-ligand concentration was increased. The most striking resemblance between both systems is the U-shaped pattern when looking at the luciferase activity relative to PEG ligand length. We speculate that at least two effects influence the formulation's efficacy and their interplay leads to the observed pattern. First, if the distance between core oligomer and folic acid is too short, an effective interaction between folic acid and its receptor will be hampered, effectively decreasing the efficacy of formulations with short PEG chains. It has also been suggested that folate receptors need to be crosslinked to facilitate uptake of nanoparticles (*Mayor, Rothberg & Maxfield, 1994*). Second, polyplexes usually lose their internal stability with increasing PEG length. This could be the reason behind the decrease in transfection efficacy with polyplexes with longer PEG chains.

All in all, results of this study suggest that lipid anchors could serve as a tool to investigate structure activity relationships on a wide variety of core polyplexes, especially when core oligomers lack functionalities for covalently binding additional structures.

Application of microfluidic devices for various tasks (*Whitesides, 2006*) and especially for producing delivery systems (*Liu et al., 2017*) usually improves quality of products. For example, *Abstiens & Goepferich (2019)* demonstrated that the continuous production of core-lipid anchor nanoparticles from PLGA and PLA-PEG with microfluidics leads to increased control over the production process which in turn generates nanoparticles with decreased size and polydispersity. Automated production of pDNA PEI polyplexes in T-junctions has been shown by *Kasper et al. (2011)* in our lab. Continuous or batch wise production of PEI pDNA polyplexes with active mixing by surface acoustic waves (SAWs) has been demonstrated by *Westerhausen et al. (2016)* and *Schnitzler et al. (2019)*. The decrease in size and polydispersity is most impressive in lipid nanoparticle formulations with siRNA (*Chen et al., 2012*; *Krzysztoń et al., 2017*). They all show that increasing mixing speeds decrease size and polydispersity. We show that similar improvements can be gained with a passive micromixer and sophisticated sequence-defined oligomers and that multi-component polyplexes can easily be prepared automatically from their starting materials.

These core-lipid anchor—PEG-ligand polyplexes were used to investigate the influence of PEG length on in vitro efficacy and to see if using lipid anchors had a predictive value for formulations with covalently bound PEG-ligands without lipid anchors. Luciferase assays revealed the influence of PEG length and PEG-ligand concentration on transfection efficiency. It has already been shown by *Klein et al. (2018)* that their shortest PEG-ligand (*P24F*) in combination with *CON* was most efficient in their study and that increasing PEG-ligand concentration increased efficiency. Additionally, they demonstrated that even if polyplexes with longer PEG-FolA ligands (*P48F*, P72F) had bound to FolA receptors, uptake is strongly decreased. They did not show, however, how PEG-ligands with shorter PEG chains compete. We strengthen the results generated by Klein et al. by showing that there is indeed a strong association between PEG length and transfection efficiency with peak performance with PEG12 and PEG24 containing polyplexes. These two are also statistically highly significant different from the core-*LA* ($p < 0.001$) or core-*LAE* ($p < 0.001$) formulation alone. In this work, however, lipid anchor oligomers with an additional PEG12 chain were used, raising the total number of PEG monomer repetitions in *LA: P12-24F* containing polyplexes to 36. All in all, our results suggest that using lipid anchors for investigating PEG-ligand performance is a valid way to screen core polyplex PEG-ligand combinations before synthesizing new structures.

Our results also suggest that transfection efficiency does not only depend on the PEG-ligand alone but on its chemical environment as well, since changing *LA* to *LAE* did significantly increase silencing efficiency of the *P12F* PEG-ligand (Fig. S23, $p < 0.001$). It did also increase silencing efficiency of *P24F* PEG-ligand on *LA* against *LAE*, albeit not significantly ($p = 0.237$). We also observed that increasing PEG length and PEG-ligand concentration increased aggregation disposition and decreased efficacy. This is in line with results from *Abstiens, Gregoritza & Goepferich (2018)* who argue that increasing PEG-ligand length and PEG-ligand concentration lead to clustering of nanoparticles and a higher probability for PEG-ligand entanglement and shrouding and therefore decreased efficacy.

The microfluidic system presented here has been designed for producing multi-component siRNA polyplexes from its starting materials. During the development process, two modules have been excessively tested: The first one to produce core polyplexes, the other one for the attachment of lipid anchors and PEG-ligands. Further development should focus on the implementation of additional modules for different task, for example for producing pDNA polyplexes. Additionally, producing polyplexes without the help of organic solvents, which possibly alters the kinetically controlled assembly process, which needs mixing speeds in the order of 50 ms (*Braun et al., 2005*) to yield small particles, could facilitate the integration of this method into acetone intolerant applications. One solution could be the utilization of SAWs (*Westerhausen et al., 2016*) to avoid usage of organic solvents. In the end, there could be a small set of modules researchers could choose from according to the desired properties of their particles. Furthermore, these modules should be integrated into a system that can automatically select from different starting materials and distribute polyplexes produced under controlled conditions to various containers. This approach would have the advantage to

enable faster production of various samples in a controlled manner while producing less waste in a shorter period of time in comparison to conventionally produced polyplexes. The advantage of high throughput production of many different formulations, albeit with a completely different system, has already been shown by *Wang et al. (2010)*.

The PEG-ligands presented here are successful in facilitating transfection when their density on the nanoparticle's surface is large enough. Research by, for example, *Lee et al. (2012)*, *Antony (1992)* and *Mayor, Rothberg & Maxfield (1994)*, show that folic acid receptors might require a certain number and distance of folic acid PEG-ligands to successfully interact with their respective nanoparticles. Therefore, this system could be employed to test various multi-folate receptors on various optimized core structures to finally get an optimized product with ideal PEG-ligands for the target cell type.

## CONCLUSIONS

In conclusion, we have shown that the controlled, continuous formulation of polyplexes from two and three components is advantageous. The subsequent anchoring of DBCO-PEGx-folic acid (PEG-ligands) coupled to lipid anchors on core polyplexes enabled us to investigate the influence of PEG length on transfection efficiency, eliminating the need to alter siRNA complexing oligomers synthetically. We found that core (*CO* + siGFP) polyplexes-lipid anchor—PEG-ligands with 12 + 12 to 12 + 24 EO repetitions had the largest silencing effect on luciferase activity in KB cells. PEG-ligands with 12 + 48 EOs, however, were prone to forming aggregates. These results were validated on a previously published system which binds PEG-ligands covalently. We confirmed that the optimal number of EO repetitions in PEG-ligands was 24. PEG-ligands with less than 24 EO repetitions are advantageous at PEG-ligand concentrations >50 mol % (relative to $n_{core\ oligomer}$), because formulations containing ≥50 mol % PEG-ligands with ≥24 EO repetitions tended to form aggregates.

## ACKNOWLEDGEMENTS

We are grateful that Sören Reinhard and Stephan Morys generated mass spectrometry data for us. We thank Wolfgang Rödl for technical support. We are deeply grateful for endless R and statistics support from Dr. Philipp Schröder. We thank the StaBLab, LMU, for reviewing statistical methods.

### Funding

This work was supported by the Deutsche Forschungsgemeinschaft (DFG), SFB1032 (projects B1 Rädler and B4 Wagner), SFB 1066 (project B5 Wagner), the Munich Center for NanoScience (CeNS), and the Cluster of Excellence Nanosystems Initiative Munich (NIM). Rafał Krzysztoń was supported by German Research Foundation (DFG) through the Graduate School of Quantitative Biosciences Munich (QBM) (GSC 1006). The funders had no role in study design, data collection and analysis, decision to publish, or preparation of the manuscript.

## Grant Disclosures

The following grant information was disclosed by the authors:
Deutsche Forschungsgemeinschaft (DFG), SFB1032 (projects B1 Rädler and B4 Wagner), SFB 1066 (project B5 Wagner), Munich Center for NanoScience (CeNS), and Cluster of Excellence Nanosystems Initiative Munich (NIM).
German Research Foundation (DFG) through the Graduate School of Quantitative Biosciences Munich (QBM) (GSC 1006).

## Competing Interests

The authors declare that they have no competing interests.

## Author Contributions

- Dominik M. Loy conceived and designed the experiments, performed the experiments, analyzed the data, contributed reagents/materials/analysis tools, prepared figures and/or tables, performed the computation work, authored or reviewed drafts of the paper, approved the final draft.
- Philipp M. Klein conceived and designed the experiments, contributed reagents/materials/analysis tools, approved the final draft.
- Rafał Krzysztoń conceived and designed the experiments, contributed reagents/materials/analysis tools, authored or reviewed drafts of the paper, approved the final draft.
- Ulrich Lächelt conceived and designed the experiments, authored or reviewed drafts of the paper, approved the final draft.
- Joachim O. Rädler authored or reviewed drafts of the paper, approved the final draft.
- Ernst Wagner conceived and designed the experiments, authored or reviewed drafts of the paper, approved the final draft.

## Data Availability

R code and data are available at Figshare: Loy, Dominik (2019): Code_raw-data. figshare. Dataset. https://doi.org/10.6084/m9.figshare.7971329.v1.

## Supplemental Information

Supplemental information for this article can be found online at http://dx.doi.org/10.7717/peerj-matsci.1#supplemental-information.

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
