# Peer review of "A microfluidic approach for sequential assembly of siRNA polyplexes with a defined structure—activity relationship"

_PeerJ Materials Science, doi:10.7717/peerj-matsci.1_

## Round 0.1 · original submission · Major Revisions

Having read the manuscript myself, I agree with the comments of the reviewers. Also, in my view, the nomenclature is confusing at times and I further missed a clear explanation of why the system shown in Fig. 1c/d is investigated in addition. Please provide some more background on this.

Reviewer 1 ·

Basic reporting

In this submission, Loy et al. describe the use of a microfluidic assembly approach to produce functionalized siRNA polyplexes. Improving control during the bottom-up manufacturing of polymer-nucleic acid particles is a worthwhile aim that could help solve the issues of batch-to-batch variability and convoluted structure-function relationships that persist. By incorporating sequence-defined oligocations and sequential chip-based assembly steps, the authors have made a nice step toward achieving fully defined polyplex micromanufacturing. The language is clear throughout, appropriate context is provided through referenced literature, and the experimental methods are both appropriate and well-described. Unfortunately, the modularity and adaptability of the system and its components is a bit of a double-edged sword - the many combinations of assembly conditions and components can cause confusion. This could be alleviated by editing some of the figures to improve clarity.

Experimental design

The methods chosen by the authors are appropriate, well-described, and well-controlled. The statistical analysis is transparent and thorough. However, there are a few issues with the results and their presentation that hinder interpretation.

1) Portions of the results are derived from nanoparticles assembled by loading ‘core polyplexes’ onto a microfluidic chip for subsequent functionalization (most of the characterization figures). Other results involve nanoparticles generated entirely on a chip from their molecular constituents (e.g. the MTT and RNAi figures). It is not clear without delving deep into the text which data is derived from which assembly paradigm. It is conceivable that different degrees of condensation could affect the ratio of available:embedded cholanic acid binding sites, thereby changing the effective molar ratio of lipid anchors and PEG ligands provided. Ideally, the characterization assays and functional assessments would be done with identical batches of particles. At the very least, the figures and captions should clearly indicate which assembly method was used in each specific case. Without such disambiguation, the results can neither be accurately interpreted nor reproduced.

2) Figure 1 is extremely confusing. It should be redrawn with descriptive labels in the figure itself instead of the caption only. Many of the acronyms used are only explained much later in the text. The rationale for including LA and LAE in the study is not discussed until Line 474, nearly halfway through the manuscript. Abbreviations like DP48F are not immediately obvious. Figure 1B should be changed to clarify that it illustrates two different assembly schemes. It also lists “Shell-ligand” where everywhere else it is called “Lipid anchor-ligand”. The abstract refers to optimized PEG lengths of 36, but the methods and Figure 1 refer only to 12, 24, 48, etc. It is not clarified until much later that this sum includes both the lipid anchor and PEG ligand. An unambiguous system should be adopted early and used consistently throughout. Fig 1C & D could be moved elsewhere in the text, perhaps nearer to the sections describing the results using the CON assembly system.

Validity of the findings

The adequate sample sizes and acceptable variability of the data enable robust statistical comparison. The trends identified by the authors are generally supported by the results. A few additional controls would strengthen some of the structure-function relationships identified.

3) It should be confirmed that the physicochemical characterizations are performed on stabilized and equilibrated particles. If size and charge continue to change over time, comparisons between samples are less reliable.

4) It is not stated whether the 500 ng dose/well was optimized. With zero toxicity observed in most conditions, it would be interesting to see a dose escalation test. Not only would this perhaps enable more robust gene silencing, but it would elucidate any differences in cytotoxicity between formulations. It would also be nice to see a validation of FA activity following microfluidic assembly.

5) The authors should speculate on the reasons a U-shape effect is observed in gene silencing relative to PEG ligand length.

Additional comments

Minor Issues:
- A MALDI analysis of the CO starting material could help validate its synthesis and monodispersity.
- Consider representing hydrodynamic diameter as Dz instead of Dh, as it is derived from the intensity-weighted Z-average.
- Figures scales should be adjusted to fit the data (e.g. Figure 2, Figure 3A).
- The asterisks/lines in Figure 5 are ambiguous. Consider using alternate symbols or tick marks to indicate significance between groups.
- The methods provided for the gel shift experiments indicate 4uL of loading dye was used, but the sample volume is not stated.
- Channel heights should be included in Figures S10 and S11
- Cell source is not provided. Are the KB cells derived in-house or obtained commercially?
- Consider using the more-standard PDI instead of pdi
- The role of acetone in the synthesis should be clarified - is it to retard siRNA compaction (line 524) or facilitate lipid anchor deposition (line 476)?
- Consider adding densitometry analysis to the gel results, as the reader is asked to compare relatively minor differences in band intensity

Typos:
Line 77: _ _
Line 97: oppositional
Line 229: hepes should be capitalized
Line 367: to-scale
Line 464: lipdid
Line 494: and
Line 296/301/305: well(s) is more standard than pocket(s)

Reviewer 2 ·

Basic reporting

This manuscript is well written. There is a nice general introduction of the topic and the results section begins with a first section in which the function of the different part of each compound is well explained and made accessible to non-specialist. All important information are given making possible to reproduce the work and possibly to use the described technology. Note that some figures and nomenclature are misleading, and I strongly advise to revise the manuscript to take this into account (see my General comments to the authors).

Experimental design

Overall, the experiments are well conducted. My only main concern was about the decorrelation between the siRNA used in the ex vivo study and the reporter monitored. Indeed, it is not clear for me why the authors measure luciferase activity whereas they used an siRNA targeting GFP. Indeed, if measuring luciferase activity, it would have made more sense to treat cells with siRNA targeting Luciferase gene; or if targeting GFP gene, why not measuring GFP fluorescence? This is a minor comment, but it would be great if the authors justify their choice?
Methods section is in general well written and complete. Especially schematic of the microfluidic devices are given as supporting material which is essential for being able to reproduce the results and use the technology. Yet, couple of minor (yet important) information are missing and should be added (see General comments below).

Validity of the findings

The conclusions and explanations proposed are in line with the results obtained from the experiments. Overall, I agree with all the conclusions exposed by the authors. Moreover, when necessary results were confirmed by an independent approach (e.g. Microscopy and FRET; electrophoresis and Ethidium Bromide assay…).

Additional comments

In their manuscript, Loy et al present a new class a transfection agent made of three blocks together with the possibility of using microfluidics to generate homogeneous nanoparticles made two to three components and characterized by controlled structure and composition. The originality of these reagents is the use of a lipid layer to self-assemble the particle through non-covalent interaction. Moreover, the authors introduce a microfluidic device allowing for preparing multilayer nanoparticles of controlled size and dispersity made of three different components in a single step. This is an elegant application of microfluidics and I’m pretty supportive to this work. However, prior to accepting this manuscript for being published in PeerJ-Materials Science, I would recommend the following points to be addressed.

Major points
1. Perhaps the biggest issue with this paper is the lack of consistency between the figures and the text regarding compounds naming. For instance, on Line 562-565 the authors talk about LA-DP48F but the reader needs to understand that this corresponds to P60F (12 Eos from LA + 48 Eos from DPF) on the figure. I acknowledge that an explanation is given in Lines 491-493, but still I have been unsure several times about this point. Another example is on line 692 where the authors state that “…reached their base at DP24F…”. Again, the reader has to understand that, if I’m correct, the authors are not talking about the bars labelled P24F, but rather P36F (ie 24 EOs from the PEG-ligand + 12 EOs from the anchor). If my understanding of the discrepancy between the text and the figure is correct, I would definitely ask the authors to change one of them and use a single nomenclature throughout their manuscript and figure.

2. I have been confused with the figures (Figure 4A and S15) showing FRET results. Indeed, in the captions, the authors say that “FRET: excites Atto488 (485 nm), measures CY5 (680 nm). This is misleading for me because we do not know (unless digging the information from the Method section) if this set of wavelengths was apply only to FRET or to all the measurement. Therefore, I would propose that the authors up-date the figures and add next to the name of the experiment (e.g. Atto488) the set of wavelengths they used for this particular measurement (in this case “Atto488 (ex./em.; 485-20 nm / 535-25 nm)”).

3. The authors mention several times “conventionally prepared core”. Even though, methods section informs that this corresponds to manually prepared nanoparticles, I think it would be wise that clarify this point the first time the sentence is used in the result section (probably line 540).

Minor points
1. In general, I wondered how efficient the transfection agents used in this work are with respect to other commercial ones frequently used for cell transfection (e.g. lipofectamine and Fugene). Do the authors have any clue on that? Such a benchmarking might be of great interest for biologist readers and would allow to better justify these new developments.

2. Line 94: The authors state that PRINT method has only advantages. If so, why searching for alternatives? It might be useful to cite disadvantages/limitations of PRINT as well.

3. Line 360: could the author give the supplier of the fluoro-silane they used?

4. Lines 363-364: the authors use a plasma treatment to bind their PDMS device onto glass slides. It is known that the gas conditions as well as the type of plasma cleaner are important as well. Can the authors indicate if the plasma was generated from air or from oxygen? Also, model and supplier of the plasma cleaner should be given.

5. Line 517: the authors mention that “additional efforts” are required in case 25% acetone is used. Could they specify what these efforts correspond to?

6. Lines 523-524: acetone was used to retard siRNA compaction. Could the authors shortly explain why this is necessary and how it works?

7. Lines 571-572: “CO in HBG with and without 50% acetone… LA or LAE with and without 50% acetone… with and without their respective PEG-ligands…”. From this sentence I was expecting 16 conditions to be tested. However, it turns out that only 8 conditions are found on Fig S14. Therefore, either data are missing or the reader should understand “CO combined with LA or LAE with and without 50% acetone… with and without their respective PEG-ligands…”, in which case the sentence should be corrected.

8. On figure S13.A, third panel, the point at 10 mL/hr indicates a size reliably twice higher than the other points. Do the authors have an explanation to propose for this striking behavior?

9. Figure S17. It might be useful for a broad readership if the authors could explain how the Ethidium Bromide displacement assay they used works.

10. Transfection assessment. I found odd that the authors used an siRNA targeting GFP mRNA and monitor Luciferase activity as a readout. Why not directly quantifying GFP fluorescence or using siRNA targeting Luciferase mRNA? I think an explanation should be given.

11. It is argued that Glutamic acids (E) were included in LAE to increase attachment. Yet, this point is not addressed anymore in the final discussion of the article which left the reader with an open question on “how useful was finally the addition of Es”. I would suggest that the authors had a statement on this point in their discussion.

12. Line 815: The authors say that T-junction is best for large scale preparation. How about their microfluidics? Could this technology also be used for large scale preparation?

13. In the text, or at least in the caption of figure 1, it would be great to explain what HBG stands for and why using this particular buffer.

Typos:
Line 123: “for or for” should read “for”
Line 675: repeated “in”
Caption Fig. 1: “lipdid” should read “lipid”

---

## Round 0.2 · accepted · Accept

The comments of the reviewers and myself have been addressed in this revised version. The manuscript has improved in readability as a result of the new nomenclature of compounds and the revised Figures 1 and 6.

# Reviewer 1 ·

Basic reporting

On the whole, the authors do a nice job of using a chemically-defined delivery system and controlled assembly to demonstrate structure-function relationships. The characterization is comprehensive, using a host of relevant methods (DLS, TEM, FRET, heparin challenge, etc.). Combined with the thorough and transparent analysis of cytotoxicity and luminescence silencing efficiency, the work will represent a nice addition to the field. I am comfortable with its publication in the current form.

Experimental design

With this resubmission, the authors have clarified murky portions of the text and included a number of new experimental results that give further validation and context to their findings. The revisions resolve the bulk of the ambiguities that existed in the previous version. In particular, the confusion regarding the covalently-attached PEG ligands (CON conditions) has been resolved by separating its discussion and creating Figure 6.

Validity of the findings

No comment

Additional comments

Minor suggestions: The author’s may wish to explain their selection of folic acid as a targeting ligand for readers outside the delivery field (e.g. potentially upregulated in cancer cells, implicated in endocytosis).